# A Framework for Inference Inspired by Human Memory Mechanisms

**Xiangyu Zeng, Jie Lin✉, Piao Hu, Ruizheng Huang, Zhicheng Zhang**
Laboratory of Intelligent Collaborative Computing,
School of Computer Science and Engineering,
University of Electronic Science and Technology of China, Chengdu, China
`{zengxy,hupiao,huangrz,zhangzc}@std.uestc.edu.cn,`
`linjie@uestc.edu.cn`

## Abstract

How humans and machines make sense of current inputs for relation reasoning and question-answering while putting the perceived information into context of our past memories, has been a challenging conundrum in cognitive science and artificial intelligence. Inspired by human brain's memory system and cognitive architectures, we propose a PMI framework that consists of perception, memory and inference components. Notably, the memory module comprises working and long-term memory, with the latter endowed with a higher-order structure to retain extensive and complex relational knowledge and experience. Through a differentiable competitive write access, current perceptions update working memory, which is later merged with long-term memory via outer product associations, reducing information conflicts and averting memory overflow. In the inference module, relevant information is retrieved from two separate memory origins and associatively integrated to attain a more comprehensive and precise interpretation of current perceptions. We exploratively apply our PMI to improve prevailing Transformers and CNN models on question-answering tasks like bAbI-20k and Sort-of-CLEVR datasets, as well as detecting equilateral triangles, language modeling and image classification tasks, and in each case, our PMI enhancements consistently outshine their original counterparts significantly. Visualization analyses reveal that relational memory consolidation, along with the interaction and integration of information from diverse memory sources, substantially contributes to the model effectiveness on inference tasks.

## 1 Introduction

Cognitive science, neuroscience and artificial intelligence (AI) collectively advance our grasp of intelligence, defined as the general mental abilities of perception, memory and reasoning, each with a unique role in human cognition. To construct more human-like intelligent systems, often referred to as the standard model of the mind (Laird et al., 2017), it is imperative to delve into the interactions among perception, memory and reasoning in a unified system. Recently, scholars have uncovered a significant flaw in previous deep learning architectures: the absence of dedicated memory module that is critical for long-term information retention and relational reasoning. This drawback becomes evident when considering the constraints of many intelligent systems, which either exclusively concentrate on perception and reasoning or intricately interweave computation with implicit memory. Therefore, many memory-based studies have emerged, mainly focusing on designing item-based memory models with recurrent neural networks (RNNs) (Hopfield, 1982; Hochreiter & Schmidhuber, 1997; Dai et al., 2019; Ramsauer et al., 2020; Schlag et al., 2021) and memory-augmented neural networks (MANNs) (Graves et al., 2014; 2016a; Le et al., 2018; Liang et al., 2023).

Nonetheless, existing approaches expose four limitations: (*i*) Implicit memory (hidden state) may gradually lose previous information as the model constantly updates its weights to accommodate new inputs, which prevents reusing the precomputed relations in sequential tasks (Vaswani et al., 2017; Santoro et al., 2017; Devlin et al., 2018). (*ii*) The memory system is configured in one of two forms: either as a singular memory unit without hierarchical construction or as multiple separate

memory components with identical data structures, both of which struggle to align with human memory traits and achieve robust generalization (Goyal et al., 2022; Dai et al., 2019; Jaegle et al., 2021; Wu et al., 2022; Kang et al., 2023; Liang et al., 2023). (*iii*) The memory-memory relation is either crude, expressed as weighted summation via neural networks or dot product attention, or it undergoes intricate memory transformation algorithms. (Vaswani et al., 2017; Santoro et al., 2018). (*iv*) Memory exploitation is confined to rudimentary retrieval, whether it's content-based addressing (Wu et al., 2020; Goyal et al., 2022; Kang et al., 2023) or explicit address (Graves et al., 2016b; Liang et al., 2023). Arguably, modern MANNs have yet to develop general architectural frameworks for learning both diverse memory components and how they should interact internally and externally.

Multiple Memory Systems Theory (MMS) asserts that working memory (WM) and long-term memory (LTM) stand as pivotal subassemblies of human cognitive processes (Atkinson & Shiffrin, 1968; Baddeley & Hitch, 1974; Eichenbaum & Cohen, 2004) , where the former serves to temporarily buffer and process data for current tasks, while the latter is responsible for the retention of relational knowledge and experiences. Additionally, the Global Workspace Theory (GWT) (Baars, 1993; Dehaene et al., 2021) suggests a communication and coordination scheme, in which disparate cognitive units write information into a shared workspace that is broadcast to all modules, along with the notion that write access is restricted.

Inspired by the MMS, GWT and cognitive theories, we assume that optimizing the structure of memory module and its internal and external correspondence mechanisms holds great promise in surmounting the extant restrictions. Accordingly, we hypothesize a cognitive framework called PMI that consists of perception, memory and inference modules, wherein memory is posited as a dual-layer memory block featuring distinct inner and outer communion principles[1]. More concretely, structurally, WM exists separately from LTM (especially the relational/declarative memory in LTM), with the latter possessing a higher-order structure to preserve intricate patterns and relations (Ryan et al., 2000; Blumenfeld & Ranganath, 2006). Regarding interactions, there are two exterior procedures: perception-based competitive writing and inference-oriented information retrieval, alongside one inner channel—designed to establish heterogeneous associations among the two memory units to facilitate efficient information filtering, storage and relational knowledge consolidation. We apply modern different neural network models like Transformers attention-based (Vaswani et al., 2017; Brown et al., 2020) and convolutional networks (He et al., 2016), which all equipped with our dual-memory module, to multifarious tasks that may require both WM and LTM: text and visual question-answering, detecting equilateral triangles, language modeling and image classification. Across all these tasks, models integrated with our memory block consistently outperform their original counterparts.

## 2 METHOD

### 2.1 OVERVIEW

An overview of our PMI framework is illustrated in Fig. 1a, which contains three pivotal components: perception, memory and inference (both potentially learned). Given an input $X$ (e.g., text, an image, or an audio signal), it is processed through a series of computational stages indexed by $t$ to derive the cognitive **u**nderstanding $U$ of the current perception, as outlined below:

1. *P* component: (Perception) — Convert the incoming input $X$ to an internal feature representation $H = \mathcal{P}(X)$.
2. *M* component: (Memory) — Update old memories given the input representation $H^{t-1}$: $M^t = \mathcal{M}(H^{t-1}, M^{t-1})$.
3. *I* component: (Inference) — Reason (interpret) the current content given the updated memories: $U = \mathcal{I}(H^{t-1}, M^t)$.

In this framework, trainable parameters are learned through backpropagation, while memory blocks are updated solely through the feedforward, which constitutes the process of memory precipitation through multiple iterations. A detailed methodological description follows.

---

[1]Code is available at https://github.com/zengxyyu/PMI-TR.

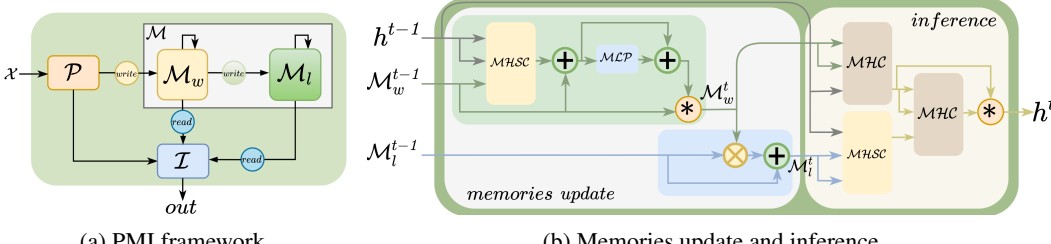

(a) PMI framework                    (b) Memories update and inference

Figure 1: Framework overview and the process of grasping the current input at calculation step $t$. (a) The memory module consists of WM $M_w$ and LTM $M_l$, each characterized by distinct data structures. (b) WM is updated by current perception via a differentiable and constrained write access, which is then integrated into LTM through outer product association. The inference component retrieves pertinent data from both WM and LTM using content-based addressing MHC and MHSC, respectively. Subsequently, through integration steps, it consolidates info from these sources to generate fresh insight into the input, which is used for next rounds of inference or to directly support the decision-making process.

## 2.2 PERCEPTION

The perceptual operation maps the original input data to internal entity representations. Focusing on the prevailing models and taking Transformers as an example, the text input undergoes embedding and positional encoding to yield initial feature representation $h^0 \in \mathbb{R}^{T \times D}$, where $T$ is the sequence length of $D$ dimension. In Vision Transformer (ViT) (Dosovitskiy et al., 2020), a $2D$ image $x \in \mathbb{R}^{H \times W \times C}$ is split into $N$ patches $x_p \in \mathbb{R}^{N \times (P^2 \cdot C)}$, each of which is linearly embedded. Then positional embeddings are added to obtain the final embedding vector $h_0 = [\, x_{class}\,;\, x_p^1\, E\,;\, x_p^2\, E\,;\, \cdots$ $;\, x_p^N\, E]+ E_{pos}\,$, $E \in \mathbb{R}^{(P^2 \cdot C) \times D}$, $E_{pos} \in \mathbb{R}^{(N+1) \times D}$, which contains fundamental feature information of the image, such as color and texture, where $(H, W)$ is the resolution of the original image, $C$ is the number of channels and $(P, P)$ is the resolution of each image patch. This process resembles the human perceptual system that receives external information and converts it into understandable internal representations, laying a foundation for subsequent memory and reasoning.

## 2.3 GLOBAL SHARED DUAL-LEVEL MEMORY

This section provides a detailed exposition of the proposed dual-layer memory module and the internal and external communication mechanisms utilized for its update, as illustrated in Fig. 1b. We posit that the memory module should be globally shared, including working memory $M_w$, which temporarily stores and processes information required for the current task, and long-term memory $M_l$ for the encoding of higher-order relational knowledge and experiences primarily in its declarative/relational memory. We opt for $M_w \in \mathbb{R}^{N \times D_m}$ as a form of WM indexed based on $m_i$ slot. Here, $N$ is the number of slots, each with a dimension of $D_m$. While the LTM is represented as a $3D$ structure $M_l \in \mathbb{R}^{C \times N \times D_m}$, with $C$ denoting memory fragments, which is biologically plausible (Marr & Thach, 1991).

### 2.3.1 EXTERNAL CHANNEL: $\mathcal{M}_w$-Write

External communication serves to update the contents of WM via two pivotal steps: competitive writing and forgetting, which informed by a fundamental aspect of the human memory system — our inclination not to retain a permanent record of every perception but rather to discerningly preserve essential elements within memory. Collectively, these processes guarantee the storage of the most critical information pertinent to the ongoing task, an indispensable facet in tasks involving reasoning.

***Write with competition*** This process aims to selectively inscribe perceived inputs into $M_w$ with finite capacity, also inspired by Miller's Law, which states that the number of information units that human WM can handle simultaneously is limited, often around $7 \pm 2$. We use a multi-headed sparse cross-attention mechanism (MHSC) for this execution, as expressed in Eq. 1, 2. Cognate to the MH mechanism used in Transformers, but MHSC exhibits two distinctive aspects: (*i*) it necessitates

separate origins for Q and K and (*ii*) it introduces a sparsity-inducing operation on the attention weight matrix. Specifically, the result of the $t - 1$ step $h^{t-1} \in \mathbb{R}^{T \times D}$ is projected into keys and values, along with $M_w^{t-1} \in \mathbb{R}^{N \times D_m}$ that is projected into queries. The current inputs compete to write through our MHSC, in conjunction with some other operations to yield the intermediate state $\widetilde{M}_w^t$. The whole formulas are as follows:

$$s_k = softmax\left(\frac{M_w^{t-1}W^Q(h^{t-1}W^K)^T}{\sqrt{d^K}}\right) \tag{1}$$

$$\widetilde{M}_w^t = s_k^* h^{t-1} W^V \tag{2}$$

$$\widetilde{M}_w^t = LN_1(\widetilde{M}_w^t + M_w^{t-1}) \tag{3}$$

$$\widetilde{M}_{w,i}^t = ReLU(MLP_i(\widetilde{M}_{w,i-1}^t)), \quad i \in \{1, \ldots, k\} \tag{4}$$

$$\widetilde{M}_w^t = LN_2(M_w^{t-1} + \widetilde{M}_{w,k}^t) \tag{5}$$

It's noteworthy that the input needs to be linearly projected to the same dimension $D_m$ as $M_w^{t-1}$ (following the traditional practice of $D = D_m$). $W^Q$, $W^K$ and $W^V$ are weight matrices. $s_k \in \mathbb{R}^{N \times (T+N)}$ is the attention weight scores of $M_w^{t-1}$ and $h^{t-1}$. Unlike the standard soft competition, we use a top-k softmax (Ke et al., 2018) to select a fixed number of entities for updating the $M_w$. $s_k^*$ denotes the post-softmax value, please consult Algorithm 1 for details. $LN_1$ and $LN_2$ signify different LayerNorms, employed to uphold memory stability over prolonged time steps. $ReLU$ is the $ReLU$ function, $MLP_i$ is the $i^{th}$ multilayer perceptron and $\widetilde{M}_{w,k}^t$ is the intermediate output through $k$ multilayer perceptrons.

***Forgetting*** Memory forgetting entails the elimination or reduction of previously stored data to make space for new info, optimizing memory performance. It is reasonable to adopt the gating mechanism since it emulates the biological memory process and effectively alleviates information conflicts. This is implemented in Eq. 6, where $I_t$ and $F_t$ indicate the input and forget gates respectively, as proposed in RMC (Santoro et al., 2018). Further details can be found in Appendix D.1.

$$M_w^t = F_t(M_w^{t-1}, h^{t-1}) \odot M_w^{t-1} + I_t(M_w^{t-1}, h^{t-1}) \odot \widetilde{M}_w^t \tag{6}$$

### 2.3.2 INTERNAL CHANNEL: $\mathcal{M}_l$-*Write*

The internal channel is utilized to update LTM that boasts a larger capacity to accommodate more relational info. As illustrated in Eq. 7, we conduct an outer product calculation between the updated $M_w^t$ and the previous-step LTM $M_l^{t-1} \in \mathbb{R}^{C \times N \times D_m}$ to merge novel vital info into the current LTM $M_l^t$. In contrast to scalar product computation that only yields a numerical value, the outer product operation (Smolensky, 1990; Halford et al., 1998) is used to capture relations and interactions between vectors, which not only enhances higher-order representational capacity but also contributes to information precipitation and memory reinforcement.

$$M_l^t = LN_3\left((M_w^t \otimes M_l^{t-1}) + M_l^{t-1}\right) \tag{7}$$

Here, $LN_3$ denotes LayerNorm, and $\otimes$ signifies the outer product operation.

### 2.4 INFERENCE

Inference component, guided by the updated memories, provides insights of current perceptions. Our interpretation of the inference is that it stems from an assumption on the form of the joint distribution between perceptual inputs and current memories. To mimic human-like ability to focus on crucial details of the ongoing task while leveraging extensive knowledge and experience to navigate complex situations, we use content-based addressing MHC that is equivalent to MHSC without sparsity and MHSC to retrieve relevant memories from $M_w^t$ and $M_l^t$ based on current input $h^{t-1}$, getting $h_w^t$ and $h_l^t$ respectively, as shown in Eq. 8-10.

$$U_w^t = MHC\left(h^{t-1}\widetilde{W}^Q, M_w^t\widetilde{W}^K, M_w^t\widetilde{W}^V\right) \tag{8}$$

$$\widehat{M}_l^t = \frac{1}{C}\sum_{i=1}^{C} M_l^t[i, :, :] \quad where \quad \widehat{M}_l^t \in \mathbb{R}^{N \times D_m} \tag{9}$$

$$U_l^t = MHSC\left(h^{t-1}\widehat{W}^Q, \widehat{M}_l^t\widehat{W}^K, \widehat{M}_l^t\widehat{W}^V\right) \tag{10}$$

Subsequently, the understanding $U_l^t$ from LTM serves to further revise and supplement the understanding $U_w^t$ from WM via the MHC mechanism, where $U_l^t$ creates queries that match with keys and values from $U_w^t$ to generate a richer representation $U_{wl}^t$. Then a linear combination of $U_w^t$ and $U_{wl}^t$ is conducted with a hyper-parameter $\alpha$ to yield the final cognition $U^t$, as shown in Eq. 12. This process of multiple correlation and fusion of various information sources contributes to extracting richer and more valuable insights that support higher-level decision-making and reasoning.

$$U_{wl}^t = MHC\left(U_l^t \bar{W}^Q, U_w^t \bar{W}^K, U_w^t \bar{W}^V\right) \tag{11}$$

$$U^t = \alpha U_w^t + (1-\alpha)U_{wl}^t \tag{12}$$

## 3   RELATED WORK

**Cognitive Science** In cognitive neuroscience, memory studies endeavor to unravel the intricacies of information storage, organization and retrieval in brains, and their profound impact on thinking, cognition and behavior, building on the pioneering work of Ebbinghaus (1885) and Bartlett & Bartlett (1932). Afterwards, Atkinson & Shiffrin (1968) proposed a multi-store model including sensory, short-term and LTM, which contributes to our insights of different memory types and stages. The successor Baddeley & Hitch (1974) further refined and delineated this model by substituting short-term memory with WM—a transient storage that can interact with LTM. Sigma (Rosenbloom et al., 2016) and Soar (Laird, 2019) are canonical cognitive frameworks of recent advancements, both of which employ a similar memory system comprising WM and LTM that play crucial roles in complex reasoning and problem-solving tasks. Moreover, the Global Workspace Theory (Baars, 1993) put forward a coordination and collaboration mechanism with restricted write access, which sheds light on the interaction of diverse cognitive components.

**Memory networks** Semi-parametric MANNs, as a form of using implicit knowledge to perform complex reasoning tasks, are a persistent theme in neural network research. Today MANNs typically rely on explicit memory and attention mechanisms, with pioneering models like Memory Networks (Weston et al., 2014) and Neural Turing Machines (NTMs) (Graves et al., 2014), both of which are equipped with a storage for vector representations accessible via attention. Memory Networks use addressable memory to execute tasks through a series of read operations. In contrast, NTMs also utilize addressable content storage, but unlike Memory Networks, which pre-load memories using all the inputs, NTMs write and read the memory one input at a time. Following this are Differentiable Neural Computers (DNC) (Graves et al., 2016a) and Sparse DNC (Rae et al., 2016), which are realized as recurrent neural networks (RNNs) capable of read and write operations on memory over time and are trained via BPTT (Werbos, 1990). A parallel research path involves enhancing RNNs like LSTM by incorporating data structures such as lists, stacks or queues (Joulin & Mikolov, 2015; Grefenstette et al., 2015).

**Transformers with memory extensions** Memory is a topic of active exploration in diverse Transformer studies. Transformer-XL (Dai et al., 2019) and its successors, Compressive Transformer (Rae et al., 2019), RMT (Bulatov et al., 2022) and Scaling Transformer (Bulatov et al., 2023) re-introduce the notion of memory and recurrence by caching self-attention hidden states from each layer into a fixed-size queue and reusing them in subsequent attention computations, with the difference that Compressive Transformer utilizes a compression network to further compress its memories into fewer vectors. In addition, various forms of global representations are introduced as a model memory that learns to gather information from input sequence tokens. Notable examples of these approaches include Set Transformers (Lee et al., 2019), ETC (Ainslie et al., 2020), Longformer (Beltagy et al., 2020) and TR+HSW (Goyal et al., 2022), all of which redesign the self-attention mechanism to reduce computational complexity. Memory modules, with their read-write global memory operations, have recently attracted attention for their potential to remember prior information, driving a movement towards more structured models. For instance, Memformer (Wu et al., 2020) proposes a dedicated external dynamic memory module after the primitive self-attention layer and interacts with it through memory reader and writer components to store previous hidden states in concise representations for efficient sequence modeling. More recently, DT-Mem (Kang et al., 2023) introduces a WM that contains N memory slots between the Transformer module and the MLP to store and retrieve information through an attention-based approach, where the Transformer module is similar to the GPT-2 (Radford et al., 2019) module without the feedforward layer. Most pertinent to our work, Goyal et al. (2022), taking cues from the GWT theory, replace Transformers' pairwise

interactions with a shared workspace featuring constrained write access—a concept equivalent to our WM that can read and write. While these endeavors are closely related to explicit memories, their memory structures are monolithic, which leads to boundaries in representing certain higher-order information or relations. Hence, one takeaway from our work is that it may be prospective to revisit previous memory enhancement methods in light of insights from cognitive science into memory structures.

# 4 EXPERIMENTS

To assess the efficacy of the PMI module in discovering and learning inferring entities and their relations, we conduct a preliminary exploration by incorporating it as a replacement for the pairwise self-attention layers in Transformers and ViT (Dosovitskiy et al., 2020), where memory components are shared globally. This modified architecture, called PMI-TR, are then applied to a diverse range of tasks, including visual QA, text-based QA, detecting equilateral triangles and language modeling. Readers can refer to Appendices E and F for the model hyperparameter settings and detailed descriptions of each task, respectively.

## 4.1 RELATIONAL REASONING : SORT-OF-CLEVR

Sort-of-CLEVR (Santoro et al., 2017) is a dataset similar to CLEVR, designed specifically for research on relational reasoning. Each 2D image in Sort-of-CLEVR is of size $75 \times 75$ and comes with 6 randomly placed geometric shapes of 6 possible colors and 2 possible shapes. There are 10 non-relational and 20 relational questions that are equally divided into binary and ternary types per image, along with corresponding answers (details in Appendix F.4). Given the bounded answer space, this task is treated as a classification task. Each image is partitioned into a sequence of uniform patches and then encoded as in ViT. Subsequently, we concatenate the image embedding with its corresponding question embedding as input into our PMI-TR, in line with Goyal et al. (2022).

For this task we evaluated our PMI-TR with the following five baselines: Standard Transformers with shared parameters across layers [TR] (Vaswani et al., 2017), Set transformer [ISAB]: Transformers where self-attention is replaced by ISAB module (Lee et al., 2019), Transformers with Shared Workspace with top-k competition [TR+HSW] (Goyal et al., 2022) and High Capacity Transformers [TR+HC]: Standard Transformers with different parameters across layers.

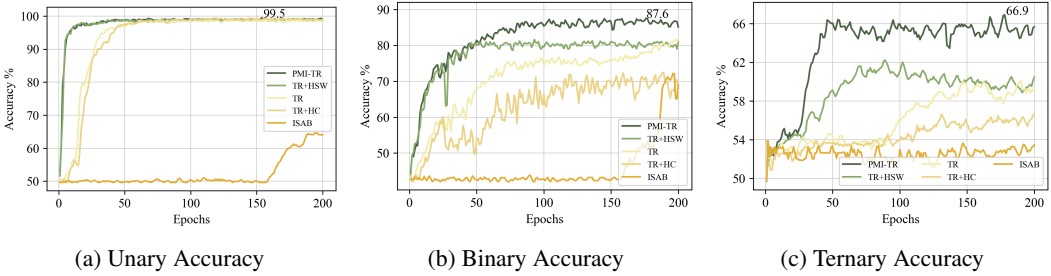

| (a) Unary Accuracy | (b) Binary Accuracy | (c) Ternary Accuracy |

Figure 2: Test accuracy vs training iterations for the Sort-of-CLEVR task.

The test accuracy curves over 200 training epochs are illustrated in Fig. 2. We observe that Transformers equipped with our global shared memory module converges faster compared to all other baselines, demonstrating superior performance on both relational and non-relational tasks. In contrast, the TR+HSW model excels in addressing non-relational questions but struggles with relational problems. We conjecture this might be because non-relational problems frequently demand the model to tackle only small amounts of information about individual objects. Interestingly, the single-level memory slots in the global workspace (similar to our WM) possess the capability to store and process scant present information, allowing them to handle these issues with ease. However, relational questions regularly necessitate multi-step reasoning to obtain an answer, such as extracting object attributes followed by relational analysis. The introduction of LTM enables the model to deposit crucial information during the learning process. Consequently, it can retrieve pertinent knowledge from this memory module in upcoming reasoning steps, going beyond its reliance

solely on the current input. This contributes to a more comprehensive understanding and handling of relational questions.

## 4.2 TEXT-BASED QA : BABI

BAbI is a pure text-based QA dataset (Weston et al., 2015) that is widely used to assess the ability of MANNs, attention mechanisms and other types of models to remember and reason on textual information. This dataset contains 20 challenging tasks, each corresponding to a particular type of reasoning, such as logical deduction, counting, pathfinding and induction, all of which possibly require both WM and LTM. Each question is associated with a set of supporting facts. For example, the facts "John journeyed to the office" and "John left the milk" support the question "Where is the milk?" (answer: "office") (more in Appendix F.2). Following Le et al. (2020b), each story is preprocessed into a sentence-level sequence, which is fed into our PMI-TR model as the input sequence. A model succeeds on a task if its performance surpasses 95%. We compare our model with recent memory networks and report the results in Table 1 (more in Appendix G.2).

Table 1: Test error rates: mean ± std. (in %) on the 20 bAbI tasks for models trained using 10k examples and best error over 10 runs. [†] is reported from Dehghani et al. (2018)

| Model | Error | |
|---|---|---|
| | Mean | Best |
| LSTM (Hochreiter & Schmidhuber, 1997) | 27.3±0.8 | 25.2 |
| TR[†] (Vaswani et al., 2017) | 22.1 | N/A |
| DNC (Graves et al., 2016b) | 12.8±4.7 | 3.8 |
| H-Mem (Limbacher & Legenstein, 2020) | 10.8 | N/A |
| NUTM (Le et al., 2020a) | 5.6±1.9 | 3.3 |
| MemNet (Dou & Principe, 2023) | 5.6 | N/A |
| TR+HSW (Goyal et al., 2022) | 3.6±0.46 | 3.25 |
| **PMI-TR** (ours) | **2.55**±0.11 | **2.32** |

## 4.3 DETECTING EQUILATERAL TRIANGLES

In this binary classification task, our goal is to determine whether a 64 × 64 sized image contains an equilateral triangle composed of three randomly placed point clusters (Ahmad & Omohundro, 2009). For equilateral triangles, the midpoints of these clusters are equidistant from each other. To feed an image into our PMI-TR, we adopt the same methodology as employed in ViT (Dosovitskiy et al., 2020). Specifically, each image is divided into equally sized 4 × 4 patches, which are then utilized as distinct input positions for the PMI-TR. In order to make precise judgments, this task requires the model to adeptly comprehend and memorize the spatial relations between disparate point clusters, embodying the relative positions and distances among them. By incorporating our PMI module, with shared WM and LTM across all layers, the model can preserve decisive info concerning each point cluster for subsequent inference procedures. Moreover, the constrained capacity of WM compels the model to selectively inscribe crucial information into the memory module, which coincides favorably with the inherent sparsity intrinsic to the task.

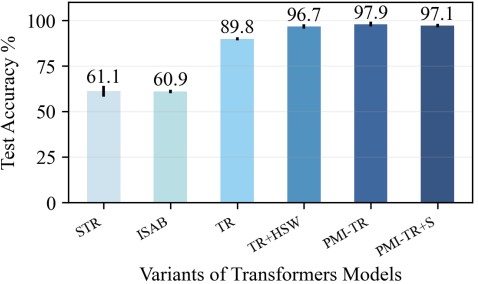

Figure 3: Detecting Equilateral Triangles. This figure compares the performance of Transformers with our PMI [PMI-TR] against other Transformer baselines.

The results in Fig. 3 reveal that PMI-TR outperforms the standard TR model in terms of both convergence speed and accuracy, converging faster (curves available in Appendix G.1) and achieving an impressive accuracy of 97.9%, an 8.1% improvement. Additionally, our approach surpasses other baselines, further confirming the efficacy of the PMI module. Here, STR denotes Transformers with sparse factorizations of the attention matrix (Child et al., 2019), [PMI-TR+S] is a variant of the PMI-TR without top-k sparsity, and other baselines are detailed in experiment 4.1.

## 4.4 LANGUAGE MODELING

To further validate the effectiveness of the proposed approach in long-sequence language modeling, we apply PMI-TR—a decoder-only transformer embedded with our PMI module, to a variety

of datasets on both character-level and word-level language modeling to have a comparison with state-of-the-art models, including Enwik8 (Matt, 2011), WikiText-103 (Merity et al., 2016) and PG-19 (Rae et al., 2019).

The experiment on Enwik8 employs a 12-layer PMI-TR (8 heads, 512 hidden size, 2048 intermediate FF), WikiText-103 uses a 16-layer PMI-TR (10 heads, 410 hidden size, 2100 intermediate FF), while PG19 uses a 12-layer PMI-TR (8 heads, 1024 embedding size, 4096 intermediate FF). The complete hyperparameter settings are available in Appendix E. The test bits-per-character (BPC) on the Enwiki8 and the perplexity (PPL) on WikiText-103 and PG-19 are reported in the Table 3, Table 2 and Table 4, respectively. Notably, we improve the results of bpc/perplexity to 0.96 on Enwik8, 16.5

Table 2: Comparison of valid and test perplexity on WikiText-103 against other models.

| Models | Layers | Params | Valid PPL | Test PPL |
|---|---|---|---|---|
| LSTM (Grave et al., 2016) | - | - | - | 48.7 |
| RMC (Santoro et al., 2018) | - | - | 30.8 | 31.6 |
| Std. Transformer-XL (Dai et al., 2019) | - | 151M | - | 24 |
| RMT (Bulatov et al., 2022) | 16 | - | - | 24.85 |
| Compressive Transf. (Rae et al., 2019) | 18 | - | 16 | 17.1 |
| Transformer-XL Large (Dai et al., 2019) | 18 | 257M | - | 18.3 |
| TIMS+HSW (Goyal et al., 2022) | 8 | 112M | 35.9 | 36.7 |
| **PMI-TR (ours)** | 8 | 116M | 24.9 | 23.8 |
| **PMI-TR (ours)** | 16 | 233M | 15.3 | **16.5** |

on WikiText-103 and 31.04 on PG19, which demonstrates the superiority of the PMI architecture. Additional qualitative analysis are available in the Appendix C.

Table 3: The test bits-per-character on Enwik8.

| Models | Layers | Params | Test BPC |
|---|---|---|---|
| Transformer-XL (Dai et al., 2019) | 12 | 41M | 1.06 |
| Transformer-XL (Dai et al., 2019) | 24 | 277M | 0.99 |
| Compressive Transf. (Rae et al., 2019) | 24 | - | 0.97 |
| Sparse Transf. (Child et al., 2019) | 30 | 95M | 0.99 |
| Adaptive Transf. (Sukhbaatar et al., 2019) | 12 | 39M | 1.02 |
| RMT (Bulatov et al., 2022) | 12 | - | 1.222 |
| TIMS+HSW (Goyal et al., 2022) | 12 | 43M | 1.36 |
| **PMI-TR (ours)** | 12 | 45M | **0.96** |

Table 4: The valid and test perplexity on PG-19.

| Models | Layers | PG19 | |
|---|---|---|---|
| | | Valid PPL | Test PPL |
| Transformer-XL (Dai et al., 2019) | 36 | 45.5 | 36.3 |
| Compressive Transf. (Rae et al., 2019) | 36 | 43.4 | 33.6 |
| $\infty$-former (Martins et al., 2021) | 12 | - | 32.48 |
| Routing Transf. (Roy et al., 2021) | 12 | - | 33.2 |
| TR+HSW (Goyal et al., 2022) | 12 | 39.46 | 32.46 |
| **PMI-TR (ours)** | 12 | 37.12 | **31.04** |

## 4.5 MORE EXPLORATIONS OF MEMORY MODULE

### 4.5.1 MEMORY ATTRIBUTES AND COMMUNICATION MODES

This section delves into qualitative analyses of memory properties and communication modes on bAbI and Sort-of-CLEVR tasks. Studies on memory properties seek to investigate how factors like capacity and persistence (global sharing) affect model performance. Experiments on communication modes aim to evaluate the efficacy of competitive writing, as well as the correction and supplementation of LTM-derived data to relevant info from WM. To tackle these questions, we set up three models of distinct sizes PMI-TR$_s$ ($l = 4, h = 4$), PMI-TR$_m$ ($l = 8, h = 8$) and PMI-TR$_l$ ($l = 12, h = 16$), and run them on various combinations of $N$, $M$ and $k$, where $l$ and $h$ are the number of layers and heads in PMI-TR, respectively, considering their critical roles in model performance.

The results are reported in Table 5, where PMI-TR$_m w/o_1$ denotes PMI-TR without memory sharing among its layers, PMI-TR$_m w/o_2$ indicates that info retrieved from LTM is directly aggregated with data from WM via $\alpha$ without correction step, and PMI-TR$_m w/o_3$ represents PMI-TR without WM involvement during inference. *soft* is a standard soft competition mode, not a top-k strategy. We can derive following key findings. For memory properties, firstly, greater memory capacity doesn't necessarily equate to better performance. The optimal results are achieved at $N = 8$ and $M = 5$, aligning with discoveries in cognitive neuroscience. Secondly, memory persistence markedly improves the performance and speed of convergence in relational inference tasks, especially in binary and ternary relations, respectively, by 7.32%, 6.88% over non-globally shared cases (more in Appendix B.1). Notably, independent memory modules result in an eightfold increase in trainable parameters. Regarding the communication mode, constrained writing exhibits heightened sensitivity in binary and ternary inference tasks, albeit with contrasting effects. We speculate that this divergence may be attributed to the larger volume of info storage required for ternary problems, thus necessitating a slightly larger $k$ value. Moreover, the impact of lacking WM is notably less than lacking LTM. Without the guidance of LTM, as an erudite scholar, there is a minor uptick in

error rate for bAbI task under the same setup, and the three types of Sort-of-CLEVR task exhibit respective decreases of 0.31% (unary), 7.34% (binary) and 4.54% (ternary) in accuracy, underscoring the constructive effect of previously accumulated relations and knowledge via outer product on relational reasoning.

Table 5: Results of ablation studies on memory properties and communication modes.

| Model | N | M | Top-$k$ | bAbI | | Sort-of-CLEVR | | | |
|---|---|---|---|---|---|---|---|---|---|
| | | | | Params | Err% | Params | Unary% | Binary% | Ternary% |
| PMI-TR$_s$ | 6 | 3 | 5 | 2.00M | 2.81 | 2.03M | 99.14 | 77.12 | 61.29 |
| | 8 | 5 | 5 | 2.27M | 2.72 | 2.29M | 99.45 | 86.06 | 62.85 |
| | 8 | 5 | 7 | 2.27M | 2.73 | 2.29M | **99.50** | 82.84 | 64.35 |
| | 10 | 7 | 9 | 2.53M | 2.78 | 2.55M | 99.19 | 80.24 | 59.48 |
| PMI-TR$_m$ | 6 | 3 | 5 | 2.07M | 2.61 | 2.09M | 99.40 | 80.13 | 65.93 |
| | 8 | 5 | 5 | 2.33M | **2.55** | 2.36M | 99.34 | **87.61** | 62.45 |
| | 8 | 5 | 7 | 2.27M | 2.57 | 2.36M | 99.19 | 81.93 | 60.89 |
| | 10 | 7 | 9 | 2.59M | 2.62 | 2.62M | 99.40 | 80.18 | 65.83 |
| PMI-TR$_l$ | 6 | 3 | 5 | 2.20M | 2.73 | 2.22M | 99.40 | 81.92 | 64.21 |
| | 8 | 5 | 5 | 2.46M | 2.58 | 2.49M | 99.14 | 84.73 | 65.52 |
| | 8 | 5 | 7 | 2.46M | 2.59 | 2.49M | 99.29 | 80.68 | **66.94** |
| | 10 | 7 | 9 | 2.73M | 2.71 | 2.75M | 99.09 | 81.47 | 65.01 |
| PMI-TR$_m w/o_1$ | 8 | 5 | 5 | 16.48M | 2.84 | 16.5M | 99.14 | 80.29 | 60.06 |
| PMI-TR$_m w/o_2$ | 8 | 5 | 5 | 2.33M | 2.91 | 2.36M | 99.19 | 79.96 | 62.40 |
| PMI-TR$_m w/o_3$ | 8 | 5 | 5 | 2.33M | 2.64 | 2.36M | 99.26 | 84.72 | 61.86 |
| PMI-TR$_m$ | 8 | 5 | $soft$ | 2.33M | 2.75 | 2.36M | 99.15 | 79.64 | 61.87 |

### 4.5.2 VISUALIZATIONS OF ATTENTION PATTERNS

To explore whether knowledge accumulates in LTM, we use visualizations of attention patterns between current perceptions and LTM on the bAbI task, shown in Fig. 4. Here, current inputs act as queries, and the LTM matrix serves as keys and values for cross-attention computation. As the depth increases, a clear trend emerges in the heatmaps: more colored regions appear that gradually stabilize and resemble, implying a growing correlation between inputs and LTM that evolvingly converges (more explanations in Appendix B.2). This may indicate that richer knowledge is accumulated in LTM, leading to a more consistent grasp of different elements within the input data across these layers.

## 5 CONCLUSION

Inspired by multiple memory systems and global workspace theory in cognitive neuroscience, we propose the PMI module containing perception, dual-layer memory and inference components, to generate a more comprehensive and accurate understanding of current inputs while depositing vital contents into memory to cope with more complex situations. We have explored PMI's dual utility: as an alternative to self-attention layers in Transformers and as a complement to convolutional networks. Extensive experiments on images and texts provide compelling evidence for the effectiveness and adaptability of PMI, meaning it could be a core component in diverse architectures. We look forward to a broader application of our method, including its integration into various frameworks and its extension to a wide range of tasks across varying modalities.

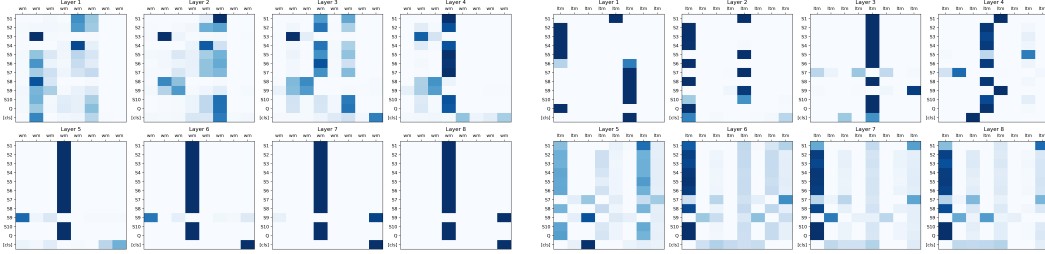

(a) Attention patterns between inputs and WM      (b) Attention patterns between inputs and LTM

Figure 4: Attention patterns between perceptions and memories across different layers of the PMI-TR.

## ACKNOWLEDGEMENTS

This work was supported by Sichuan Province Science and Technology Support Program, No.:2022ZHCG0008.

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

## Appendix

## A  PSEUDO CODES

---

**Algorithm 1:** PMI-TR Algorithm

---

***Notation:*** Given an original input sequence of length $T$ with embedding dimension of $D$ for the proposed model. Let $h^l$ denotes the output of the $l^{th}$ layer. The working memory is represented as a matrix $M_w \in \mathbb{R}^{N \times D_m}$ with distinct compartmentalized memories for each row, where $m_{w,i}$ is the state of slot $i$ (the total number of slots is $N$). The long-term memory is represented as $3D$ matrix $M_l \in \mathbb{R}^{C \times N \times D_m}$, where $C$ is the number of memory segments.

***Initialization:*** Convert the raw input $X \in \mathbb{R}^{T \times vocab\_size}$ to $h_0 = p_x \oplus e_x \oplus cls_{token}$, where $h_0 \in \mathbb{R}^{T \times D}$, $p_x$ signifies the positional encoding, $e_x$ corresponds to the embedding of $X$ and $cls_{token}$ is the classification head. Initialize the working memory matrix $M_w$ and the long-term memory matrix $M_l$, respectively, guaranteeing their universal sharing across all layers encompassed within our framework.

***Input to the layer l:*** $h^{l-1}$ having shape $\mathbb{R}^{T \times D}$

***Step 1: The current perceptions compete with each other to be selected to update the $M_w$***

- $Q = M_w^{l-1} W^q$

- $s_k = softmax\left(\frac{Q\left(h^{l-1}W^k\right)^T}{\sqrt{d^k}}\right)$, where $s_k \in R^{N \times T}$

- Construct a set $\mathcal{F}_t$ containing the indices of the $k$ selected specialists with the top-$k$ largest values of $s_k$.

  $s_k^* = \begin{cases} s_{[n,t]}, & t \in \mathcal{F}_t, \\ 0, & t \notin \mathcal{F}_t \end{cases}$   for all $n \in \{1, \ldots, N\}, t \in \{1, \ldots, T\}$

***Step 2: External communication: $\mathcal{M}_w$-Write***

- $residual = M_w^{l-1}$

- $\widetilde{M}_w^l = LN_1(s_k^* h^{l-1} W^v + residual)$

- $\widetilde{M}_{w,i}^l = \text{ReLU}\left(\text{MLP}_i(\widetilde{M}_{w,i-1}^l)\right)$   $i \in \{1, \ldots, k\}$, $k$ signifies the number of layers in the MLP.

- $\widetilde{M}_w^l = LN_2(\widetilde{M}_{w,k}^l + residual)$

- $M_w^l = F_l(M_w^{l-1}, h^{l-1}) \odot M_w^{l-1} + I_l(M_w^{l-1}, h^{l-1}) \odot \tanh(\widetilde{M}_w^l)$
  $F_l$ refers to the forget gate and $I_l$ represents the input gate of $l^{th}$ layer.

***Step 3: Internal communication: $\mathcal{M}_l$-Write with the updated $M_w$***

- $M_l^l = LN_3((M_w^l \otimes M_l^{l-1}) + M_l^{l-1})$

***Step 4: Make sense of the current perceptions based on the updated $M_w$ and $M_l$: $\mathcal{M}_w$-Read and $\mathcal{M}_l$-Read***

- $h_w^l = softmax\left(\frac{h^{l-1}\widetilde{W}^q\left(M_w^l \widetilde{W}^k\right)^T}{\sqrt{d^k}}\right) M_w^l \widetilde{W}^v$

- $\widehat{M}_l^l = \frac{1}{C} \sum_{i=1}^C M_l^l[i,:,:]$   where $\widehat{M}_l^l \in \mathbb{R}^{N \times D_m}$

- $c_k = softmax\left(\frac{h^{l-1}\widehat{W}^q\left(\widehat{M}_l^l \widehat{W}^k\right)^T}{\sqrt{d^k}}\right)$

- $h_l^l = c_k^* \widehat{v}$   where   $\widehat{v} \in \widehat{M}_l^l \widehat{W}^v$
  Similar to $s_k^*$, $c_k^*$ represents the outcome of the top-$k$ largest sparse attention scores associated with $c_k$.

- $h_{wl}^l = MHC\left(h_l^l \bar{W}^Q, h_w^l \bar{W}^K, h_w^l \bar{W}^V\right)$

- $h^l = \alpha h_w^l + (1-\alpha)h_{wl}^l$

---

## B  Experiments on Memory Consolidation

### B.1  More Ablation Results

In this section, we present the results of ablation experiments concerning the global sharing of memory components. The plot in Fig. 5 illustrates the change in testing accuracy over training iterations. Here, the PMI-TR model denotes the usage of a globally shared working memory module, while PMI-TR$_{w}/o$ signifies the absence of global sharing. TR represents the vanilla Transformers model. The results demonstrate several key findings. Firstly, across all three tasks, both our PMI-TR and PMI-TR$_{w}/o$ models exhibit faster convergence compared to the traditional TR model. Secondly, when compared to the baselines TR and PMI-TR$_{w/o}$, PMI-TR achieves the highest accuracy and convergence rate, particularly in complex binary and ternary relation tasks, showcasing the effectiveness of the adopted global sharing strategy in complex relational reasoning tasks. However, for non-relational tasks, their performance remains relatively on par, potentially due to the fact that non-relational tasks often emphasize extracting direct patterns and information from input data, which coincidentally aligns with the effective capture of these correlations by the self-attention mechanism in Transformers.

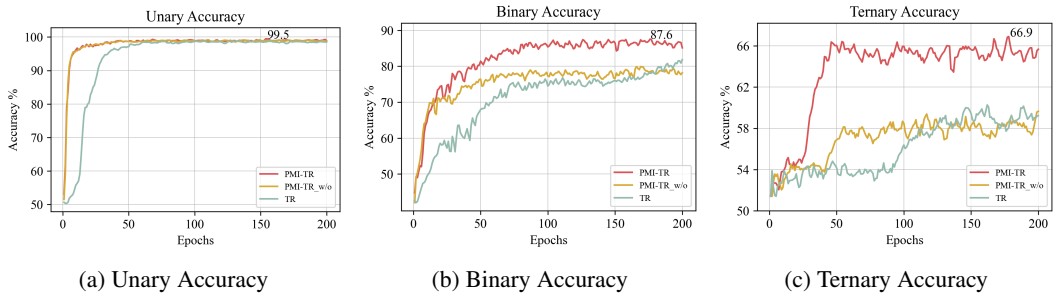

|                    |                    |                    |
| :----------------: | :----------------: | :----------------: |
| (a) Unary Accuracy | (b) Binary Accuracy | (c) Ternary Accuracy |

Figure 5: Test accuracy vs. training iterations for the Sort-of-CLEVR task. Results of the ablation experiments on the memory module's global sharing or persistence.

### B.2  Visualization

We present four Q&A instances from the bAbI dataset, illustrating the attention matrix of perception towards both working memory (Fig. 6a) and long-term memory (Fig. 6b) during the reasoning process within eight layers of the PMI-TR model. Each row corresponds to distinct Q&A examples, with columns representing the layers of our PMI-TR. Moreover, in each heatmap, the vertical axis is a representation of perceptual information, functioning as queries, while the horizontal axis refers to long-term memory or working memory, serving as keys in the cross-attention mechanism.

As an illustration, let's consider a specific problem with its set of stories and corresponding answer. In the first line of the working memory attention patterns, each $Si$ represents a distinct sentence in the narrative. For instance, $S1$ corresponds to 'the hallway is east of the bathroom', and $S2$ corresponds to 'the bedroom is west of the bathroom', and so forth until $S_{max}$, where the maximum $S_{max}$ for each task may vary (in task 1, $S_{max}$ is S10). In cases where there are fewer than $S_{max}$ story sentences, placeholder sentences (all zeros) are used as filler input. The letter 'Q' represents the question: 'What is the bathroom east of?', and 'CLS' is the classification header.

Firstly, it is a well-known fact that the magnitude of attention weights is reflected through color intensity, where the deeper the color, the higher the weight. Figure 6a illustrates attention patterns between perception and working memory across various layers of PMI-TR for four instances from the bAbI dataset. During the inference process, with an increase in layers within the PMI-TR model, a clear pattern emerges in the heatmaps: the presence of colored blocks markedly diminishes, yet the intensity of color within the remaining blocks steadily deepens. This implies a growing focus of working memory on essential segments of current inputs, especially those indispensable for question-answering. This result is consistent with the restrained write access elucidated in Section 2.3.1, which encourages working memory to concentrate on the utmost critical elements of the task.

Figure 6b depicts attention patterns between perception and long-term memory across different layers. For each problem (per row), as layers increase (from left to right), the heat maps show a clear trend opposite to the working memory pattern, with more and more colored areas but less intensity, and the last several heat maps tend to be stable and approximate. This suggests that long-term memory gradually accumulates more knowledge as computation step $t$ increases, and thus exhibits moderate correlation with a broader input area (unlike working memory with limited capacity, which focuses only on the most important info). Similar attention patterns in the final layers imply convergence of the three-dimensional long-term memory matrix within finite calculation steps, resulting in a more consistent understanding of various input aspects.

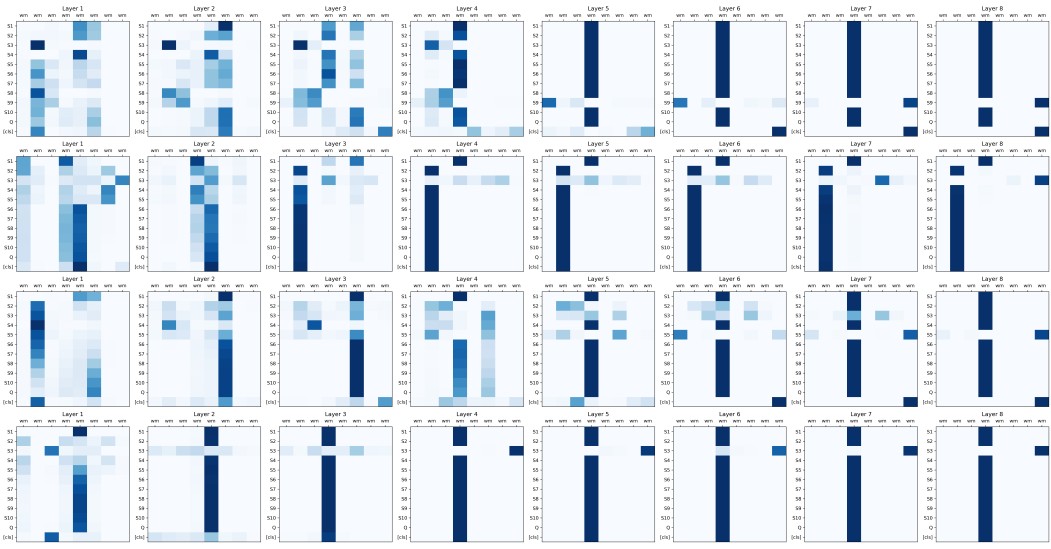

(a) Attention patterns between perception and working memory across different layers of the PMI-TR for 4 examples from the bAbI dataset.

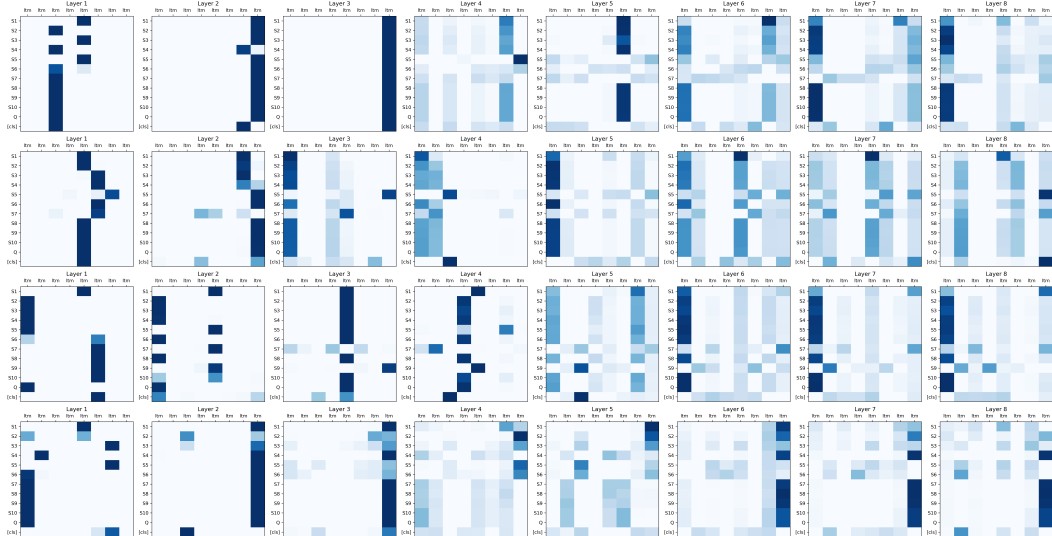

(b) Attention patterns between perception and long-term memory across different layers of the PMI-TR for 4 examples from the bAbI dataset.

Figure 6: Attention patterns between perception and memory.

### B.3  IMAGE CLASSIFICATION : CIFAR-10

In order to establish the generality of the PMI framework, we extend its application beyond PMI-TR mentioned above to include a convolutional series model and evaluate its performance on the CIFAR-10 dataset. CIFAR-10 is a benchmark image dataset commonly used in the field of computer vision, which consists of 50k training and 10k test images of resolution $32 \times 32$ with a total of 10 classes. Specifically, the original images, after four convolutional layers, serve as perceptions into our PMI module to obtain understandings, which then undergo linear and softmax transformations to yield final classification results.

The performance of the best models on test sets is reported in Table 6, where CNN_PMI w/o refers to CNN_PMI without guidance from LTM. It's obvious that both PMI-TR and CNN_PMI models exhibit superior performance, achieving accuracies of 79.12% and 78.69% in CIFAR-10, respectively, with an improvement of 2.94% (compared to TR) and 0.08% (compared to CNN_MLP). These results further underscore the universality of our PMI module.

Table 6: Results of different models on CIFAR-10.

| Models | Trans. | | | | Conv. | | |
|---|---|---|---|---|---|---|---|
| | ViT | ISAB | TR+HSW | PMI-TR (ours) | CNN_MLP | CNN_PMI (ours) | CNN_PMI w/o (ours) |
| Acc (%) | 76.18 | 76.39 | 76.28 | **79.12** | 78.61 | **78.69** | 78.63 |
| Params (M) | 0.75 | 2.21 | 2.01 | 2.0 | 0.11 | 1.75 | 1.68 |

## C  QUALITATIVE RESULTS

For a qualitative analysis of working memory and long-term memory in long-term language modeling, we illustrate an instance of the attention histogram between different memory types and the current input sequence during the inference stage on the WikiText-103 dataset when predicting the next token. The attention histogram for long-term memory in Figure 7 highlights its ability to focus on more long-range and relevant information, significantly exceeding the span of working memory depicted in Figure 8.

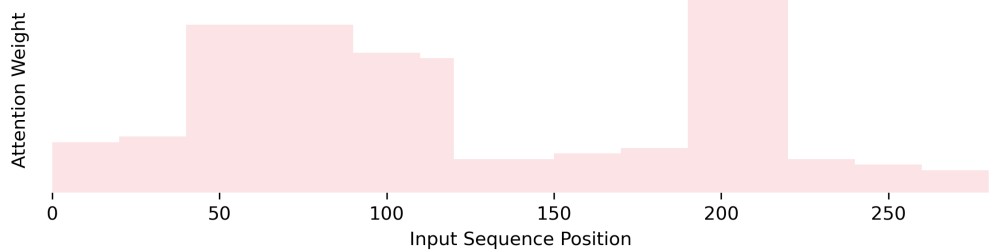

Figure 7: Attention histogram of long-term memory to the current input sequence during inference in PMI-TR, when predicting the ground truth word "Japanese". The words in the LTM which receive higher attention ($>0.5$) are shaded.

**Ground Truth**: Although they had lost contact during the night , the Americans did find the Japanese

Shortages of aircraft and serviceability problems greatly retarded pilot training and the ships only had a total of 17 D4Ys and 18 <unk>on hand on 1 October ; of these , only 6 and 16 were operational , respectively . The Japanese plan for the defence of the Philippines envisioned that the surviving carriers would be used to lure the American carrier forces away from the invasion area to a position where the carriers could be attacked by land @-@ based aircraft and the transports by the rest of the IJN . The other carrier air groups were not in much better shape and

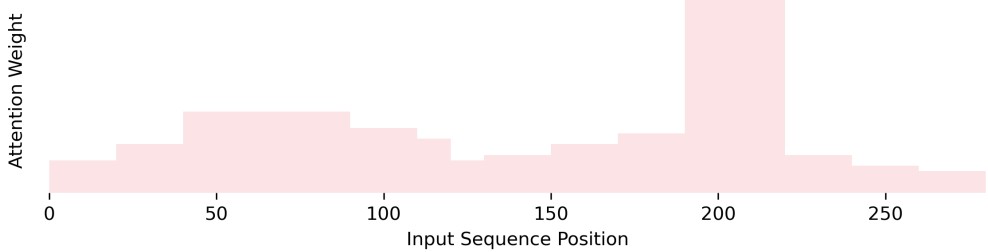

Figure 8: Attention histogram of working memory to the current input sequence during inference in PMI-TR, when predicting the ground truth word "Japanese". The words in the WM which receive higher attention (>0.5) are shaded.

the Japanese decided to retain the aircraft ashore for use against the American carriers . The Fourth Carrier Division was assigned to the Northern Force under the command of Vice Admiral Jisaburō Ozawa and the sisters sailed from Yashima on 20 October . On the morning of 24 October , the bulk of the few aircraft aboard were launched to attack the American carriers as a distraction . They inflicted no damage and caused the Americans to search in the direction from which they had attacked . The Americans finally spotted the Japanese carriers at 16 : 40 , some 200 miles ( 320 km ) east of Cape Engaño , the northeastern tip of Luzon . The American carriers were spread out and it was very late in the day to launch an airstrike , so Admiral William Halsey , commander of the Third Fleet decided to mass his carriers in a position to attack the following morning . Ozawa reversed course during the night , correctly believing that the Americans would follow him north .

## D IMPLEMENTATION DETAILS

### D.1 GATING MECHANISM

To update the working memory, we employ the gating mechanism introduced by Santoro et al. (2018), which consists of an input gate and a forget gate. Let $X^{t-1} = [X_1^{t-1}, X_2^{t-1}, \cdots, X_T^{t-1}] \in \mathbb{R}^{B \times T \times D}$ represent the perceptual input, and $M_w^{t-1}, M_w^t \in \mathbb{R}^{B \times N \times D_m}$ represent the previous and updated working memory, respectively. $\widetilde{M}_w^t$ is the intermediate result described by Eq. 5 in Section 2.3.1. The gating mechanism can be formulated as follows.

$$\bar{X} = \frac{1}{T} \sum_{i=1}^{T} \text{relu}(X_i \times W^I)$$
$$K = \bar{X} + \tanh(M_w^{t-1}) \times W^F$$
$$I_t = \text{sigmoid}(K + b_i)$$
$$F_t = \text{sigmoid}(K + b_f)$$
$$M_w^t = I_t \times \tanh(\widetilde{M}_w^t) + F_t \times M_w^{t-1}$$

Here, $I_t$ and $F_t$ represent the input and forget gates of the current calculation step $t$, with corresponding weight matrices $W^I$ and $W^F$. The biases for the input and forget gates are denoted as $b_i$ and $b_f$. In practice, we set $b_i = 0$, $b_f = 1$, $D = D_m$.

## E HYPERPARAMETERS SETTING

The hyperparameter settings of the PMI-TR model on all tasks are shown in Table 7 and Table 8, where Adam and AdamW were proposed by (Kingma & Ba, 2014) and (Loshchilov & Hutter,

2017), respectively. The other baseline models remain the same configuration for the corresponding tasks.

Table 7: The hyperparameter setting of PMI-TR model on four tasks.

| Parameters | Tasks | | | |
| --- | --- | --- | --- | --- |
| | bAbI | Sort-of-CLEVR | Triangle | Cifar-10 |
| Top-k | 5 | 5 | 5 | 5 |
| Number of layers | 8 | 4 | 2 | 4 |
| Number of attention heads | 8 | 4 | 4 | 4 |
| Embedding dimensions | 256 | 256 | 128 | 256 |
| Optimizer | Adam | Adam | Adam | AdamW |
| Weight decay | N/A | N/A | N/A | 0.09 |
| Learning rate | 0.0002 | 0.0001 | 0.0001 | 0.0002 |
| Batch size | 64 | 64 | 100 | 64 |
| Inp Dropout | 0.1 | 0.1 | 0.1 | 0.1 |
| Seed | 1 | 1 | 1 | 1 |
| Number of working memory slots ($N$) | 8 | 8 | 8 | 8 |
| Number of long-term memory segments ($M$) | 5 | 5 | 5 | 5 |
| Size of each working memory slot ($D_m$) | 256 | 256 | 256 | 256 |
| Size of each long-term memory segment | $5 \times 256$ | $5 \times 256$ | $5 \times 256$ | $5 \times 256$ |
| Number of MLP layers in attention | 4 | 4 | 5 | 4 |
| Memory attention heads | 8 | 4 | 1 | 1 |
| Gate style | 'unit' | 'unit' | 'unit' | 'unit' |
| Initial $\alpha$ value | 0.7 | 0.75 | 0.7 | 0.55 |

Table 8: The hyperparameter setting of PMI-TR model on three language modeling tasks.

| Parameters | Tasks | | |
| --- | --- | --- | --- |
| | Enwik8 | WikiText-103 | PG-19 |
| Top-k | 5 | 5 | 5 |
| Number of layers | 12 | 16 | 12 |
| Number of attention heads | 8 | 10 | 8 |
| Embedding dimensions | 512 | 410 | 1024 |
| Optimizer | Adam | Adam | Adam |
| Weight decay | 0.5 | 0.5 | 0.5 |
| Learning rate | 0.00025 | 0.00025 | 0.00025 |
| Batch size | 64 | 64 | 64 |
| Inp Dropout | 0.1 | 0.1 | 0.1 |
| Seed | 1 | 1 | 1 |
| Number of working memory slots ($N$) | 8 | 8 | 8 |
| Number of long-term memory segments ($M$) | 5 | 5 | 5 |
| Size of each working memory slot ($D_m$) | 512 | 410 | 1024 |
| Size of each long-term memory segment | $5 \times 512$ | $5 \times 410$ | $5 \times 1024$ |
| Number of MLP layers in attention | 4 | 4 | 5 |
| Memory attention heads | 8 | 4 | 1 |
| Gate style | 'unit' | 'unit' | 'unit' |
| Initial $\alpha$ value | 0.7 | 0.7 | 0.7 |

# F    DESCRIPTION OF THE TASKS/DATASETS

## F.1    ENWIKI8&WIKITEXT-103&PG-19

Enwiki8 (Matt, 2011) is utilized for character-level language modeling and comprises 100M bytes of unprocessed Wikipedia text, where the first 90MB for training, 5MB for validation and the latter 5MB for testing. Both WikiText-103 (Merity et al., 2016) and PG-19 (Rae et al., 2019) are benchmarks for word-level language modeling with long-term dependency, with the former containing 103M tokens from 28K English Wikipedia articles and the latter from English books published before 1919.

## F.2    BABI

The bAbI-20k dataset consists of 20 distinct text-based QA tasks, each presenting a unique reasoning challenge, including counting, deduction and induction. Each task is divided into training, validation and test datasets, with 9k, 1k and 1k questions respectively. They are presented in the form of short stories or text passages, comprising narratives, questions, answers and supporting facts. Narratives introduces entities, actions and contextual info relevant to the question, and the answer is substantiated by facts from the narratives. Four detailed examples are shown below.

---

**Task 1: Single Supporting Fact**
1. John travelled to the hallway.
2. Mary journeyed to the bathroom.
3. Daniel went back to the bathroom.
4. John moved to the bedroom.
Where is Mary? A: bathroom    S: 2

---

**Task 8: Lists/Sets**
1. Mary got the milk there.
2. John moved to the bedroom.
3. John picked up the football there.
4. John journeyed to the bathroom.
What is John carrying? A: football    S: 3

---

**Task 12: Conjunction**
1. Mary and Daniel travelled to the bathroom.
2. John and Daniel travelled to the office.
3. Sandra and Daniel moved to the kitchen.
4. Sandra and John moved to the garden.
Where is Sandra? A: garden    S: 4

---

**Task 16: Basic Induction**
1. Julius is a lion.          2. Lily is a rhino.
3. Bernhard is a swan.    4. Lily is white.
5. Bernhard is green.     6. Greg is a rhino.
7. Greg is gray.          8. Julius is white.
9. Brian is a lion.
What color is Brian? A: white    S: 9 1 8

---

## F.3    DETECTING EQUILATERAL TRIANGLES

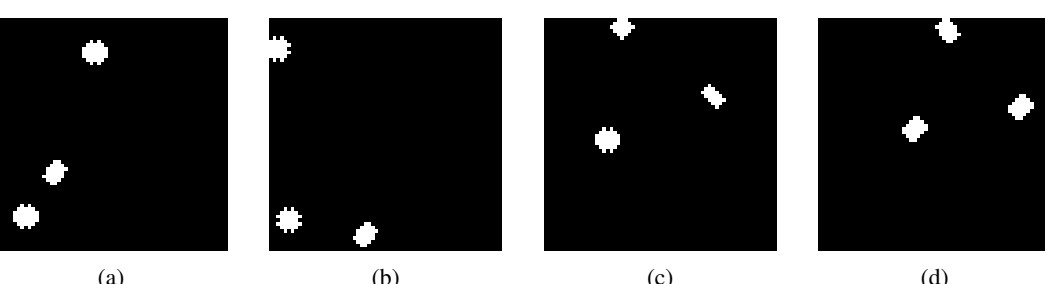

| (a) | (b) | (c) | (d) |

Figure 9: An illustration of the equilateral triangle detection task. Here, (a) and (b) refer to non-equilateral triangle, (c) and (d) refer to equilateral triangle.

## F.4  SORT-OF-CLEVR

**Ternary questions**
Q: How many objects are in the rectangle formed by the centers of the red and yellow objects?
    Answer: 1
Q: Are there any objects on the line formed by the centers of the red and grey objects?
    Answer: yes
Q: How many objects form an obtuse triangle with a blue object and a yellow object?
    Answer: 3

**Binary questions:**
Q: What is the shape of the object that is furthest from the green object?
    Answer: orange
Q: What is the color of the object that is closest to the red object?
    Answer: green
Q: How many objects have the shape of the yellow object?
    Answer: 3

**Unary questions:**
Q: What is the shape of the red object?
    Answer: circle
Q: Is the green object on the left or right of the image?
    Answer: left
Q: Is the grey object on the top or bottom of the image?
    Answer: top

Figure 4: An instance from the sort-of-clevr dataset.

# G  ADDITIONAL EXPERIMENTAL RESULTS

## G.1  TEST CURVES FOR DETECTING EQUIVALENT TRIANGLES

As shown in Fig. 10, we observe that our proposed models, PMI-TR and its variant, PMI-TR+S without top-k sparsity, show significantly faster convergence compared to other models. It is worth noting that the PMI-TR model achieves slightly higher accuracy compared to the PMI-TR+S model, highlighting the usefulness of our competitive writing mechanism.

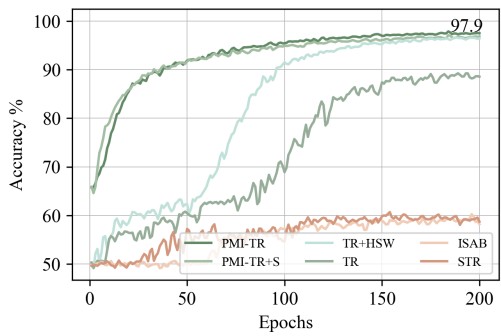

Figure 10: Comparison of convergence speeds on equilateral triangle detection task.

## G.2  BABI DETAILED RESULTS

Detailed results are shown in the Table 9.

| Task | run1 | run2 | run3 | run4 | run5 | run6 | run7 | run8 | run9 | run10 | Mean±std |
|------|------|------|------|------|------|------|------|------|------|-------|----------|
| 1:Single Supporting Fact | 0 | 0 | 0 | 0 | 0 | 0 | 0 | 0 | 0 | 0 | 0 |
| 2:Two Supporting Facts | 0.21 | 0.08 | 0 | 0.10 | 0 | 0.52 | 0.56 | 0.73 | 0.35 | 0.62 | 0.32±0.28 |
| 3:Three Supporting Facts | 1.35 | 0.35 | 0.03 | 0.15 | 0.56 | 1.88 | 0.92 | 0.04 | 0.15 | 1.19 | 0.66±0.64 |
| 4:Two Arg. Relations | 0 | 0 | 0 | 0 | 0 | 0.21 | 0 | 0 | 0.31 | 0 | 0.05±0.11 |
| 5:Three Arg. Relations | 0.52 | 0.46 | 0.52 | 0.73 | 0.52 | 0.52 | 0.73 | 0.73 | 0.83 | 0.42 | 0.60±0.14 |
| 6:Yes/No Questions | 0 | 0 | 0 | 0 | 0 | 0 | 0 | 0 | 0 | 0 | 0 |
| 7:Counting | 0.31 | 0.25 | 0.04 | 0.73 | 0.35 | 0.35 | 0.35 | 0.46 | 0.15 | 0.04 | 0.30±0.21 |
| 8:Lists/Sets | 0 | 0 | 0.21 | 0.21 | 0.31 | 0.10 | 0.52 | 0.10 | 0.10 | 0.21 | 0.18±0.16 |
| 9:Simple Negation | 0 | 0 | 0 | 0 | 0 | 0 | 0 | 0 | 0 | 0 | 0 |
| 10:Indefinite Knowledge | 0 | 0 | 0.31 | 0.10 | 0 | 0 | 0 | 0 | 0.10 | 0 | 0.05±0.10 |
| 11:Basic Coreference | 0 | 0 | 0 | 0 | 0 | 0 | 0 | 0 | 0 | 0 | 0 |
| 12:Conjunction | 0 | 0 | 0 | 0 | 0 | 0 | 0 | 0 | 0 | 0 | 0 |
| 13:Compound Coref. | 0 | 0 | 0 | 0 | 0 | 0 | 0 | 0 | 0 | 0 | 0 |
| 14:Time Reasoning | 0 | 0 | 0 | 0 | 0 | 0 | 0.10 | 0 | 0 | 0 | 0.01±0.03 |
| 15:Basic Deduction | 0 | 0 | 0.21 | 0 | 0 | 0 | 0 | 0 | 0 | 0 | 0.02±0.07 |
| 16:Basic Induction | 48.21 | 47.12 | 44.96 | 47.92 | 48.04 | 48.63 | 47.35 | 50.81 | 47.27 | 48.56 | 47.89±1.47 |
| 17:Positional Reasoning | 0.94 | 0.03 | 0.03 | 0.31 | 0.58 | 1.67 | 0.83 | 0.60 | 0.77 | 0.38 | 0.61±0.49 |
| 18:Size Reasoning | 0.42 | 0.01 | 0.00 | 0.52 | 0.21 | 0.21 | 0.21 | 0.31 | 0.52 | 0.21 | 0.26±0.18 |
| 19:Path Finding | 0.00 | 0 | 0 | 0.21 | 0 | 0 | 0.10 | 0 | 0.10 | 0 | 0.04±0.07 |
| 20:Agent's Motivations | 0 | 0 | 0 | 0 | 0 | 0 | 0 | 0 | 0 | 0 | 0 |
| Average | 2.60 | 2.42 | **2.32** | 2.55 | 2.53 | 2.70 | 2.58 | 2.69 | 2.53 | 2.58 | 2.55±0.11 |
| Failed task(>5%) | 1 | 1 | 1 | 1 | 1 | 1 | 1 | 1 | 1 | 1 | |

Table 9: Results from 10 test runs of the PMI-TR model on 20 bAbI tasks, each consisting of 10k samples, following 200 epochs of joint training. Bold denotes the best run.

