# OpenReview forum: "A Framework for Inference Inspired by Human Memory Mechanisms"
_ICLR.cc/2024/Conference — ICLR 2024 poster_

### Official Review · Reviewer_4HGz · 2023-10-24

**Soundness:** 3 good
**Presentation:** 2 fair
**Contribution:** 3 good
**Rating:** 8
**Confidence:** 4

**Summary:**

This work presents a novel architecture inspired by memory systems in cognitive science. The method improves performance across multiple reasoning tasks in both transformer and CNN architectures.

**Strengths:**

- The proposed architecture improves performance across a diverse set of reasoning tasks.
- A reasonable set of baseline comparisons are included.
- An ablation study is performed to assess the impact of specific components.

**Weaknesses:**

- The primary limitation concerns the framing of the architecture as instantiating both working memory and longterm memory. It is not clear to me that the architecture actually involves longterm memory in any meaningful sense. I think the approach would be better described as a form of relational working memory (utilizing a tensor product to capture relational information). This of course doesn't concern the method itself, which seems to perform well across multiple tasks. But I think the contribution would be much more clearly framed as a kind of working memory that exploits *relational* information. The role of relations in working memory is very well-studied in cognitive science (see references below), and I think this would make an interesting topic for discussion.
- Is it possible to study an ablation model that includes the 'longterm' memory component but not the 'working' memory? It seems likely that the tensor product in the longterm memory component is primarily driving the gain in performance, and it would be nice if this could be isolated.
- It would be good to cite work from cognitive science on the role of tensor product representations in working memory [1,2] as this is highly related to the outer product mechanism in the 'longterm' memory module.

[1] Smolensky, P. (1990). Tensor product variable binding and the representation of symbolic structures in connectionist systems. Artificial intelligence, 46(1-2), 159-216.

[2] Halford, G. S., Wilson, W. H., & Phillips, S. (1998). Processing capacity defined by relational complexity: Implications for comparative, developmental, and cognitive psychology. Behavioral and brain sciences, 21(6), 803-831.

Minor comments:
- It sounds like what is referred to as the transformer baseline in this work is actually a 'universal transformer' [3] in which parameters are shared across layers, and what is referred to as a 'high capacity transformer' is just a standard transformer (in which each layer has different parameters).

[3] Dehghani, M., Gouws, S., Vinyals, O., Uszkoreit, J., & Kaiser, Ł. (2018). Universal transformers. arXiv preprint arXiv:1807.03819.

**Questions:**

- In what sense does the 'longterm' memory module involve long term memory more than the 'working' memory module? They both seem to operate over the same timescale, the only difference being the presence of the tensor product to capture relational interactions (which is not related to longterm vs. working memory).
- Is it possible to ablate the 'working' memory module?

---

> ### Author Response · Authors · 2023-11-19
> **Relational Working Memory/ Difference between WM and LTM+ Ablation Experiment Without Working Memory+ References+ Explanation of Terms**
>
> We sincerely appreciate the reviewers' thorough reviewing of this paper and the valuable insights the reviewers have provided.
>
> **1.Relational Working Memory/ Difference between WM and LTM**
>
> We are enthused that the reviewers gained a profound understanding of this paper. We would like to provide the following clarifications to address the concerns of the reviewer. In cognitive neuroscience, working memory and long-term memory differ in the following aspects: (i) purpose: working memory is primarily employed for processing and manipulating current information, while long-term memory is more geared towards storing and retrieving extensive information, including acquired knowledge and experiences. (ii) capacity: working memory is constrained by a limited capacity, whereas long-term memory boasts a significantly larger capacity. In our research, we utilize a content-based sparse cross-attention mechanism for updating working memory, where the current perception serves as both the key and value, while working memory acts as the query. In contrast, long-term memory is primarily updated through tensor outer product operations, which are not directly tied to the current perception but rather involve extracting higher-order relationships from the cumulative content of working memory. With a larger three-dimensional tensor, long-term memory can accommodate a greater volume of information. This can be interpreted as working memory paying more attention to current perceptual information, while long-term memory involves extracting higher-order relationships from the accumulated content of all previous working memories. During our research, we have indeed reviewed numerous relevant papers the role of relations in working memory. We would like to make appropriate modifications in the next version. Thank you once more for your insightful suggestions.
>
> **2.Ablation Experiment Without Working Memory**
>
> We prefer to believe that the presence of the working memory component is justified, both from a cognitive science perspective and in the context of Transformers' attention mechanism. This is because it utilizes an attention mechanism to encode crucial information from the current perception into the working memory, which is consistent with the human cognitive process. The reviewers also acknowledge that long-term memory involves higher-order extraction of relations in working memory. If the structure of working memory were absent, it could require substantial changes to the research.
> However, to investigate whether the tensor product in long-term memory is the primary factor driving the gain in performance, we conducted an ablation experiment during the inference phase by removing the working memory (i.e., not utilizing the understanding of the current perception from the working memory, omitting formulas 8 and 11 in the paper, and directly extracting the understanding from long-term memory). The results are reported in the table below. Across various tasks, the absence of working memory led to a slight performance decline compared to the original best cases. However, this performance decline is far less than the impact of removing long-term memory, which indicates the importance of tensor outer product in long-term memory.
>
> | Tasks                              |Layers |MITR                  | MITR without WM      |
> | --------------------------------- | ----------------------- | ---------------------- |---------------------- |
> | SORT-OF-CLEVR                     |8| 99.34, 87.61, 62.45     | 99.26 , 84.72 , 61.86   |
> | bAbI                              |8| 2.55                    | 2.64                   |
> | DETECTING EQUILATERAL TRIANGLES   |8| 97.9                    | 97.2                   |
> | Enwik8  [1]                         |12| 0.96 bpc                | 1.01 bpc               |
> | WikiText-103 [2]                  |16   | 16.5 ppl                | 17.2 ppl               |
>
> Here, 99.34, 87.61 and 62.45 represent the accuracy of the unary, binary and ternary problems, respectively.
>
> **3.References**
>
> We appreciate your valuable insights, and we plan to include these references in the next revision.
>
> **4.Explanation of Terms**
>
> We would like to explain that, in this paper, transformer baseline is a standard transformer in which parameters are shared across layers, and a 'high capacity transformer' refers to a standard transformer, in which each layer has different parameters.
>
> Reference
>
> [1] Mahoney, M. (2011). Large text compression benchmark.
>
> [2] Merity, S., Xiong, C., Bradbury, J., & Socher, R. (2016). Pointer sentinel mixture models. arXiv preprint arXiv:1609.07843.

---

> > ### Comment · Reviewer_4HGz · 2023-11-22
> > **Response to rebuttal**
> >
> > Thanks very much to the authors for these responses. I have a point-by-point reply below.
> >
> > ## Long-term memory and working memory terminology
> >
> > I remain unconvinced that 'long-term memory' and 'working memory' are the appropriate terms for these modules. In cognitive psychology, long-term memory and working memory are defined primarily in terms of the timescale over which memories are maintained. While it is true that human working memory is capacity-constrained, this is not part of the definition of working memory. Therefore, I do not think it makes sense to describe one process as working memory and another as long-term memory simply because one is more capacity-constrained, when both operate over the same timescale. I also think that the contribution of the paper would be much clearer if the *relational* nature of the component referred to as 'long-term memory' were highlighted more, rather than confusingly giving the impression that these two components operate over different timescales.
> >
> > ## 'Working memory' ablation
> >
> > Thank you to the authors for carrying out an ablation of the 'working memory' module. While these results are informative, I think they imply that a more substantial revision of the paper is needed. The 'working memory' module does not appear to have any effect on performance (performance is better on some tasks and worse on others, and the differences are very small, it's unclear whether they are statistically significant). This suggests that the proposed model's performance is due entirely to the relational representation in the 'long term memory' module. In my opinion, the paper would be much clearer if the 'working memory + longterm memory' framing was abandoned, and the 'working memory' module was removed, highlighting instead the importance of using a relational memory representation for reasoning tasks.
> >
> > ## Universal transformer
> >
> > Thank you for adding the reference for universal transformers, but I still think it is confusing to describe these baselines as a 'transformer' and a 'high-capacity transformer'. The 'high-capacity transformer' is just a standard transformer, and the 'transformer' is a universal transformer. This is not clearly explained in the paper.
> >
> > Overall, I am still happy to vote for acceptance, but unless the issues regarding framing and clarity can be addressed I will stick with my current score (6) for now.

---

> > > ### Author Response · Authors · 2023-11-23
> > > **Universal Transformer+ Working Memory Ablation+ Long-term Memory and Working Memory Terminology**
> > >
> > > Dear Reviewer 4HGz,
> > >
> > > **1.Universal Transformer**
> > >
> > > We have thoroughly considered your suggestion and have made modifications to this issue in the latest reversion just uploaded. We sincerely appreciate your guidance and corrections.
> > >
> > > **2.'Working Memory' Ablation**
> > >
> > > As for the ablation results (without working memory), we concede that our previous elucidation may not have been as explicit as necessary, potentially leading to minor misinterpretations. We would like to point out that ablation results indicate that during the inference, the absence of working memory results in an overall decline in performance across all tasks compared to cases where both working memory and long-term memory are present (albeit to a significantly lesser extent than the absence of long-term memory alone). This is because for tasks such as Sort-of-Clevr and detecting equilateral triangles, higher accuracy correlates with better performance. Conversely, for bAbI, Enwik8 and WikiText-103 tasks, as measured by errors, bits-per-character (bpc) and perplexity (ppl) respectively, lower values are indicative of superior performance.
> > >
> > > **3.Long-term Memory and Working Memory Terminology**
> > >
> > > In this study, the difference between working memory and long-term memory is not solely based on distinct capacities but also on their distinct processing capabilities for information. Working memory is primarily involved in storing and processing temporary information, while long-term memory denotes a more advanced stage involving the extraction and processing of information, which, as we previously hypothesized, may include the extraction of relations within working memory.
> > > The relationship between working memory and the GWT theory is closely intertwined, supported by numerous studies [1,2]. Consequently, we believe that the depiction of the term "working memory" might be relatively clear and reasonable. Regarding the term "long-term memory", after carefully considering your suggestions and reviewing relevant literature, we propose describing it as "relation memory" for enhanced clarity. This term captures and stores certain outer product relations within working memory, a concept extensively explored in various studies [3,4,5].
> > >
> > > However, due to time constraints and the substantial textual adjustments required, we find it challenging to achieve the desired version within the limited timeframe. Nevertheless, we commit to revise the description of the term "long-term memory" in the next version, attributing the main contributions of this work to the extraction of high-order relations within working memory.
> > >
> > >
> > > We appreciate your understanding and valuable feedback. We would be more than happy to provide further details during our reviewer-author discussion. Your response is of great importance to us, and we look forward to hearing from you.
> > >
> > > Thank you sincerely!
> > >
> > > Authors
> > >
> > > **References**
> > >
> > > [1] Juliani, A., Arulkumaran, K., Sasai, S., & Kanai, R. (2022). On the link between conscious function and general intelligence in humans and machines. arXiv preprint arXiv:2204.05133.
> > >
> > > [2] Baars, B. J., & Franklin, S. (2003). How conscious experience and working memory interact. Trends in cognitive sciences, 7(4), 166-172.
> > >
> > > [3] Le, H., Tran, T., & Venkatesh, S. (2020, November). Self-attentive associative memory. In International Conference on Machine Learning (pp. 5682-5691). PMLR.
> > >
> > > [4] Kohonen, T. (2012). Self-organization and associative memory (Vol. 8). Springer Science & Business Media.
> > >
> > > [5] Ranganath, C., Cohen, M. X., Dam, C., & D'Esposito, M. (2004). Inferior temporal, prefrontal, and hippocampal contributions to visual working memory maintenance and associative memory retrieval. Journal of Neuroscience, 24(16), 3917-3925.

---

> > > > ### Comment · Reviewer_4HGz · 2023-11-23
> > > > **Followup**
> > > >
> > > > Thanks very much to the authors for these final responses and revisions. I trust that they will update the framing of the final paper in line with our discussions. I will update my score to an 8.

---

> > > > > ### Author Response · Authors · 2023-11-23
> > > > > **A Reply for Gratitude**
> > > > >
> > > > > Dear Reviewer 4HGz,
> > > > >
> > > > > We sincerely appreciate your trust and valuable feedback. Thank you very much for the time and effort you have dedicated to our research in these days. In the next reversion, we assure you that we will make the agreed-upon modifications. We are at your disposal for any additional information that you may need.
> > > > >
> > > > > Finally, we extend our warm wishes for a joyful Thanksgiving Day, and may each day be delightful for you.
> > > > >
> > > > > Best Regards,
> > > > >
> > > > > Authors

---

> ### Author Response · Authors · 2023-11-22
> **Gentle reminder: less than one day left for the author-reviewer discussion**
>
> Dear Reviewer 4HGz,
>
> We sincerely appreciate the time and consideration you dedicate to our work. In response to your insightful comments, we have made extensive improvements in the latest version (accessible at https://openreview.net/pdf?id=vBo7544jZx). We would like to know if these modifications have effectively addressed any concerns on your part. We would be more than happy to provide further details during our reviewer-author discussion. Your response is of great importance to us, and we look forward to hearing from you.
>
> Thank you sincerely!
>
> Authors

---

### Official Review · Reviewer_AVcH · 2023-11-01

**Soundness:** 3 good
**Presentation:** 3 good
**Contribution:** 3 good
**Rating:** 6
**Confidence:** 4

**Summary:**

The paper proposes a cognitive framework called PMI that consists of perception, memory, and reasoning modules. It is inspired by human memory mechanisms and aims to improve the understanding and handling of relational questions in AI systems. The memory module includes working memory (WM) and long-term memory (LTM), with LTM having a higher-order structure to retain accumulated knowledge. Current perceptions update WM through competitive write access and are merged with LTM via outer product associations. The inference module retrieves relevant information from both WM and LTM to generate comprehensive insights. The PMI enhancements consistently outperform their original counterparts in tasks such as question-answering, relation calculation, and image classification.

**Strengths:**

- Integration of cognitive science and AI: The paper draws inspiration from multiple memory systems theory and global workspace theory in cognitive neuroscience, and applies these insights to develop the PMI framework for AI systems.

- Novel memory module: The PMI framework introduces a dual-layer memory block with distinct communion principles, featuring working memory (WM) and long-term memory (LTM). This structure allows for efficient information filtering, storage, and knowledge consolidation.

- Enhanced performance: The PMI enhancements consistently outperform their original counterparts in various tasks such as question-answering, and image classification. This demonstrates the effectiveness of the proposed framework in improving AI systems' understanding and reasoning abilities.

- Clear experimental results: The paper provides detailed experimental results, including accuracy rates and convergence rates, to support the effectiveness of the PMI module. Visualizations of attention patterns further illustrate the model's ability to consolidate and integrate information from different memory sources.

- Reproducibility: The authors plan to share their code once the review process is completed, ensuring the reproducibility of their experiments and allowing for further research and development in this area.

**Weaknesses:**

- The text appears to be excessively embellished. I would like to encourage the author to employ conventional terminology, as exemplified by the authors referencing "relation calculation" in the abstract.

- The paper includes visualizations of attention patterns between perceptions and memories, but it could benefit from providing more detailed explanations and interpretations of these visualizations.

- Examining the qualitative impact of your modules on various types of tasks would provide valuable insights, rather than solely relying on quantitative results. This approach would enhance the paper's overall credibility. You can achieve this by employing various visualization techniques and similar methods.



**Additional Feedback and Future Experiments:**

- Enhance the clarity of the text to facilitate a deeper comprehension of the paper. Despite grammatical accuracy, the writing occasionally comes across as artificial

- Incorporate additional visualizations to facilitate a clearer and more easily comprehensible text.

- Consider expanding the scope of this research to include more memory-based and cognition-inspired tasks. You may find the paper titled "Decoding the Enigma: Benchmarking Humans and AIs on the Many Facets of Working Memory" by Sikarwar et al. to be a relevant reference in this context.

**Questions:**

Please see weaknesses above.

---

> ### Author Response · Authors · 2023-11-19
> **Conventional Terminology/Improve Clarity+ Explanations and Interpretations of These Visualizations+ Qualitative Impact/Incorporate Additional Visualizations+ More Memory-based and Cognition-inspired Tasks**
>
> We thank the reviewers for their time in reviewing the paper and providing constructive feedback, and sincerely appreciate the reviewers for their recognition of our approach.
>
> **1.Conventional Terminology/Improve Clarity**
>
> We sincerely appreciate the reviewer for giving us thorough feedback regarding the writing style of this paper, we will incorporate the feedback in the next revision of the paper to avoid being artificial. We will also make appropriate adjustments to the presentation structure, particularly in the visualization section, to enhance clarity.
>
> **2.Explanations and Interpretations of These Visualizations**
>
> We would like to emphasize that we do provide a more detailed explanation in Appendix B.2 (given the limited length in the main text), illustrating the meaning of visualizations with specific examples. Detailed explanations for each case can be found in the legends of Figures 6a and 6b. Taking your valuable suggestions into account, we will diligently revise this presentation style in the next revision. We plan to appropriately integrate textual explanations into both the main text and the body of the appendix, rather than presenting them solely in the form of legends. If you find the explanations insufficient, we would like to provide additional explanations and interpretations in the next revision.
>
> **3.Qualitative Impact/Incorporate Additional Visualizations**
>
> For this suggestion, we conduct a qualitative analysis of the proposed modules from a priori perspective on various types of tasks. We posit that as the computational steps increase, the content in working memory and long-term memory can serve as a prior for reasoning. Moreover, we use the bertviz library [14] to add visualizations of attention distributions for perception and memory on the bAbI dataset in the appendix. In addition, we add attention histograms for working memory and long-term memory on the WikiText-103 [2] dataset (new experiment) in the appendix. When predicting the next token, the attention histograms of these two different types are notably distinct. The attention histogram for long-term memory indicates its ability to focus on more long-range and relevant information, which is significantly longer than working memory.
>
> **4.More Memory-based and Cognition-inspired Tasks**
>
> We conduct additional experiments by adding the PMI framework into the decoder of Transformers for language modeling, covering enwik8 [1], WikiText-103 [2] and PG-19 [3], to broaden the scope of this research. The results are as follows.
>
> In addition, we thoroughly reviewed the paper "Decoding the Enigma: Benchmarking Humans and AIs on the Many Facets of Working Memory", which you kindly shared. We consider it an outstanding benchmark. In the next revision, we plan to conduct experiments on this dataset to further evaluate the performance of our PMI architecture.
>
> **Table 1**: Test set bits-per-character on enwik8.
>
> | Models| Layers|Params |BPC|
> |--|--|-|--|
> | Transformer-XL  [8]| 12 | 41M |1.06|
> | Transformer-XL  [8] | 24 | 277M | 0.99|
> | Compressive Transformer  [3]  | 24| - | 0.97|
> | Sparse Transformer [12]  | 30| 95M | 0.99|
> | Adaptive Transformer [13] |12| 39M| 1.02 |
> | RMT [9] | 12     | - | 1.222|
> | TIMS+HSW [5]*  | 12| 43M| 1.36|
> | MITR (ours) | 12| 45M| **0.96**|
>
> **Table 2**: Comparison with other models on WikiText-103.
>
> | Models | Layers | Params |Valid PPL |Test PPL |
> |--|-|-|--|--|
> | LSTM [6]|  - | - | -  | 48.7   |
> | RMC [7]|-| 30.8   | - | 31.6 |
> | Standard Transformer-XL [8] | - | 151M   | - | 24 |
> | RMT [9]  | 16|- |-  | 24.85|
> | Compressive Transformer [3] | 18| - | 16 | 17.1|
> | Transformer-XL Large [8]  | 18| 257M   | - | 18.3|
> | TIMS+HSW [5]* | 8| 112M   | 35.9 | 36.7|
> | MITR (ours)   | 8 | 116M | 24.9 | 23.8 |
> | MITR (ours)   | 16| 233M| 15.3| **16.5** |
>
> **Table 3**: Results on language modeling on PG-19 dataset.
>
> | Models | Layers|Valid PPL|Test PPL|
> |--|--|-|--|
> | Transformer-XL [8] | 36| 45.5| 36.3|
> | Compressive Transformer [3]   | 36 | 43.4| 33.6|
> | ∞-former [10]  |12| - | 32.48|
> | Routing Transformer [11] | 12| - | 33.2|
> | TR+HSW [5]* | 12 | 39.46 |32.46|
> | MITR (ours) | 12 |37.12| **31.04**|
> Here, * signifies the results obtained from our experiments.
>
> [14] https://github.com/jessevig/bertviz

---

> ### Author Response · Authors · 2023-11-22
> **Gentle reminder: less than one day left for the author-reviewer discussion**
>
> Dear Reviewer AVcH,
>
> We are grateful for your valuable feedback. In response to your insightful comments, we have made extensive improvements in the latest version (accessible at https://openreview.net/pdf?id=vBo7544jZx). Specifically, we have provided additional clarity on visualization in Appendix B.2, and introduced new findings from qualitative analysis in Appendix C.
>
> We are eager to ascertain if these modifications have effectively addressed any concerns on your part. We would be more than happy to provide further details during our reviewer-author discussion. Your response holds great importance for us, and we sincerely appreciate the time and consideration you dedicate to our work.
>
> Thank you sincerely!
>
> Authors

---

### Official Review · Reviewer_SKET · 2023-11-03

**Soundness:** 3 good
**Presentation:** 3 good
**Contribution:** 3 good
**Rating:** 6
**Confidence:** 3

**Summary:**

Inspired by human brain’s memory system and cognitive architectures,this paper propose a PMI framework that consists of perception, memory and inference components. Notably, the memory module comprises working and long-term memory, with the latter endowed with a higher-order structure to retain more accumulated knowledge and experiences.


In my opinion, the motivation of this paper is meaningful because it comes from the human brain's memory.
And the proposed memory module looks like powerful because it consists of working memory and long-term memory.
However, the experiments may not be enough due to it not compare with other memory augment models, such as the memory augment language model. and this paper not take experiments on language generative task.
Besides, the

**Strengths:**

1. The motivation is sometimes novel and comes from human's brain memory.

2. The proposed model is meaningful with its novel motivation

3. The paper is well written, and the image is easy to understand.

**Weaknesses:**

1. the experiments may not be enough to compare it with other memory-assisted language model

2. The experiments is hard to understand, and i think it is not necessary to conduct experiments on image classification. And there is little work on the memory augment image model due to the image's too long context.

3. I don't see any connection between your work and the title, the author maybe need change a title due to this model hard to help us underanding AI .

I hope the author takes more experiments on the language model to solve the longer context challenge.

**Questions:**

No

---

> ### Author Response · Authors · 2023-11-19
> **More Experiments on Language Modeling+ Compare with Other Memory Augment Models+ Image Classification+ Title**
>
> We thank the reviewers for their time in reviewing the paper and providing constructive feedback, and sincerely appreciate the reviewers for their recognition of our approach.
>
> **1.More Experiments on Language Modeling**
>
> To further validate the effectiveness of the proposed approach in long-sequence language modeling, we applied MITR—a decoder-only transformer embedded with our PMI module, to a variety of datasets on both character-level and word-level language modeling, including enwik8 [1], WikiText-103 [2] and PG-19 [3]. Enwiki8 is utilized for character-level language modeling and comprises 100M bytes of unprocessed Wikipedia text. Both WikiText-103 and PG-19 are benchmarks for word-level language modeling with long-term dependency, with the former containing 103M tokens from 28K English Wikipedia articles and the latter from English books published before 1919.
> Part of the implementation is based on Transformer-XL repository [4]. The experiment on Enwik8 employs a 12-layer MITR (8 heads, 512 hidden size, 2048 intermediate FF), WikiText-103 uses a 16-layer MITR (10 heads, 410 hidden size, 2100 intermediate FF), while PG19 uses a 12-layer MITR (8 heads, 1024 embedding size, 4096 intermediate FF). In addition, we use Adam optimizer with linear schedule learning rate starting from 0.00025 for 500,000 steps and set the working memory size M=8, the long-term memory size N=5 and top-k=5 on all the datasets. The complete hyperparameter settings are available in our repository as well as in the Appendix.
>
> The test bits per character (BPC) on the Enwiki8 dataset and the perplexity (PPL) on WikiText-103 and PG-19 are reported in the table below. Notably, we improve the results of bpc/perplexity to 0.96 on enwiki8, 16.5 on WikiText-103 and 31.04 on PG19, which demonstrates the superiority of the PMI architecture.
>
> **Table 1**: Test set bits-per-character on enwik8.
>
> | Models| Layers|Params |BPC|
> |-|-|-|-|
> | Transformer-XL  [8]|12| 41M |1.06|
> | Transformer-XL  [8] |24| 277M|0.99|
> | Compressive Transformer  [3]|24|-|0.97|
> | Sparse Transformer [12]| 30| 95M |0.99|
> | Adaptive Transformer [13] |12| 39M|1.02 |
> | RMT [9] |12| - |1.222|
> | TIMS+HSW [5]*|12| 43M|1.36|
> | MITR (ours) |12|45M| **0.96**|
>
> **Table 2**: Comparison with other models on WikiText-103.
>
> |Models|Layers|Params|Valid PPL|Test PPL|
> |-|-|-|-|-|
> | LSTM [6]|-|-|-|48.7|
> | RMC [7]|-| 30.8| -|31.6 |
> | Standard Transformer-XL [8] |-|151M|-| 24 |
> | RMT [9]|16|-|-| 24.85|
> | Compressive Transformer [3] |18|-|16|17.1|
> | Transformer-XL Large [8]| 18| 257M| - |18.3|
> | TIMS+HSW [5]* | 8| 112M| 35.9 |36.7|
> | MITR (ours) |8|116M |24.9|23.8|
> | MITR (ours) |16| 233M|15.3|**16.5**|
>
> **Table 3**: Results on language modeling on PG-19 dataset.
>
> | Models | Layers|Valid PPL|Test PPL|
> |-|-|-|-|
> | Transformer-XL [8]|36|45.5| 36.3|
> | Compressive Transformer [3]| 36 |43.4| 33.6|
> | ∞-former [10]|12|-|32.48|
> | Routing Transformer [11] |12|-| 33.2|
> | TR+HSW [5]*|12 |39.46 |32.46|
> | MITR (ours) |12|37.12|**31.04**|
> Here, * signifies the results obtained from our experiments.
>
> **2.Compare with Other Memory Augment Models**
>
> We would like to point out that we do provide comparisons with other memory-augmented models. For example, TR+HSW model mentioned in this paper is a variant of Transformers with a working memory proposed by Goyal et al. [5]. In the text-based question-answering bAbI in Section 4.2 (Table 1), we compare with various memory-augmented models, including DNC, NUTM and H-Mem, etc. Moreover, in the newly introduced language modeling tasks, we compare with several memory-augmented models, such as LSTM [6], RMC [7], Transformers-XL [8], and Compressive Transformer [3], etc.
>
> **3.Image Classification**
>
> For image classification, our aim is to demonstrate that the architecture embedded with our PMI, whether it belongs to the transformers or convolutional series, could outperform the original models. This is to establish the generality of the PMI, showcasing its applicability beyond Transformers. Additionally, the original intent of the paper is to prove that the introduction of PMI could enhance the model’s inference ability, which we think maybe include the ability for image classification, while not approaching it from the perspective of long sequences. However, considering your constructive suggestion, we will place this experiment in the appendix, while adding language modeling experiments into the main text. What are your thoughts on this arrangement?
>
> **4.Title**
>
> After thoughtful consideration of your valuable feedback, we are planning to revise the title to "A PMI Framework for Inference Inspired by Human Memory Mechanisms", considering that the focal point of this paper lies in leveraging theoretical insights from cognitive science to construct a cognitive architecture named PMI, and embed it into mainstream architecture such as Transformers and convolutional networks to help us build AI systems for inference that align more closely with human cognition.

---

> > ### Author Response · Authors · 2023-11-19
> > **Corresponding References**
> >
> > [1] Mahoney, M. (2011). Large text compression benchmark.
> >
> > [2] Merity, S., Xiong, C., Bradbury, J., & Socher, R. (2016). Pointer sentinel mixture models. arXiv preprint arXiv:1609.07843.
> >
> > [3] Rae, J. W., Potapenko, A., Jayakumar, S. M., & Lillicrap, T. P. (2019). Compressive transformers for long-range sequence modelling. arXiv preprint arXiv:1911.05507.
> >
> > [4] https://github.com/kimiyoung/transformer-xl
> >
> > [5] Goyal, A., Didolkar, A., Lamb, A., Badola, K., Ke, N. R., Rahaman, N., ... & Bengio, Y. (2021). Coordination among neural modules through a shared global workspace. arXiv preprint arXiv:2103.01197.
> >
> > [6] Grave, E., Joulin, A., & Usunier, N. (2016). Improving neural language models with a continuous cache. arXiv preprint arXiv:1612.04426.
> >
> > [7] Santoro, A., Faulkner, R., Raposo, D., Rae, J., Chrzanowski, M., Weber, T., ... & Lillicrap, T. (2018). Relational recurrent neural networks. Advances in neural information processing systems, 31.
> >
> > [8] Dai, Z., Yang, Z., Yang, Y., Carbonell, J., Le, Q. V., & Salakhutdinov, R. (2019). Transformer-xl: Attentive language models beyond a fixed-length context. arXiv preprint arXiv:1901.02860.
> >
> > [9] Bulatov, A., Kuratov, Y., & Burtsev, M. (2022). Recurrent memory transformer. Advances in Neural Information Processing Systems, 35, 11079-11091.
> >
> > [10] Martins, P. H., Marinho, Z., & Martins, A. F. (2021). $\infty $-former: Infinite Memory Transformer. arXiv preprint arXiv:2109.00301.
> >
> > [11] Roy, A., Saffar, M., Vaswani, A., & Grangier, D. (2021). Efficient content-based sparse attention with routing transformers. Transactions of the Association for Computational Linguistics, 9, 53-68.
> >
> > [12] Child, R., Gray, S., Radford, A., & Sutskever, I. (2019). Generating long sequences with sparse transformers. arXiv preprint arXiv:1904.10509.
> >
> > [13] Sukhbaatar, S., Grave, E., Bojanowski, P., & Joulin, A. (2019). Adaptive attention span in transformers. arXiv preprint arXiv:1905.07799.

---

> > ### Comment · Reviewer_SKET · 2023-12-05
> > **Sorry for your waiting.**
> >
> > Thanks for your detailed response.
> >
> > Based on your new experimental result, I'll improve my score from 5 to 6.
> >
> > Meanwhile, I'm willing to see that as a more suitable title.
> >
> > Further, a recently published paper may be related to your work, in which they also get inspiration from human memory.
> >
> >
> > [1] Few-shot Generation via Recalling Brain-Inspired Episodic-Semantic Memory.

---

> ### Author Response · Authors · 2023-11-22
> **Gentle reminder: less than one day left for the author-reviewer discussion**
>
> Dear Reviewer SKET,
>
> We would like to express our gratitude for your valuable feedback. In the latest revision (available at https://openreview.net/pdf?id=vBo7544jZx), we have diligently addressed each of the concerns you raised. We are eager to learn if these changes have effectively alleviated any confusion on your part and more than willing to provide further details during the author-reviewer discussion. Your response is immensely important to us, and we genuinely appreciate the time and consideration you dedicate to our work.
>
> Thank you sincerely!
>
> Authors

---

> ### Author Response · Authors · 2023-11-23
> **Gentle reminder: less than six hours left for the reviewer-author discussion**
>
> Dear reviewer SKET,
>
> We sincerely apologize for taking up some of your time, and we greatly appreciate the time and effort you have dedicated to reviewing our work. However, we've noticed that the summary part of your initial comments appears to have been truncated ("Besides, the……"), and we wonder if it's due to a system issue or other reasons. We genuinely hope to receive your complete comments.
>
> Furthermore, we have individually addressed the other issues you raised, visible to us in the latest reversion (accessible at https://openreview.net/pdf?id=vBo7544jZx). We are looking forward to your reply and at your disposal for any additional information.
>
> Finally, we wish you a happy Thanksgiving Day and may each day be delightful for you.
>
> Best Regards,
>
> Authors

---

### Meta-Review · Area_Chair_33Lr · 2023-12-06

**Metareview:**

This is an interesting paper with fairly novel ideas, and reviewers found the strengths to be robust evaluation on diverse tasks, reasonable baselines and ablations, clear results and good motivation.

Reviewers initially had valid concerns about framing, articulationg and presentation, as well as certain aspects of the evaluations. However, the authors reponded with extensive additional results/analyses, detailed explanations and some writing improvements (including a title change). Ultimately, reviewers were unanimous that the work met the standard for acceptance to ICLR. This AC personally likes the motivation and approach of the paper, as well as the diverse evaluation tasks.

Overall, there is agreement to recommend this work for acceptance.

[  With that said, I very strongly urge the authors to improve the paper further with the following points about writing/presentation, in order for the work to achieve better readability and ultimately stronger impact:
-- Avoid using acronyms in the title
-- Carefully re-think whether naming/coining both "PMI" and "MITR" differently is absolutely necessary. Too many new terms adds to difficulty in understanding by the reader.
-- Reduce the terminology and make it consistent. The papers uses "framework", "module", "architecture" and "model".
-- Simplify the words and phrases used (e.g. "distinct communion principles of the inner and outer")
-- Double-check for typos, etc. (e.g. "Appendix E F")

Thanks,
AC  ]

**Justification For Why Not Higher Score:**

The presentation leaves much to be desired, and more convincing articulations/illustrations and qualitative analyses would have made for a much stronger paper.

**Justification For Why Not Lower Score:**

Interesting ideas coupled with robust evaluations and good results.

---

### Decision · Program_Chairs · 2024-01-16

Accept (poster)